# Unlearning with Asymmetric Sources: Improved Unlearning-Utility Trade-off with Public Data

**Ahmed Mehdi Inane** [1 2]  **Vincent Quirion** [1 2]  **Gintare Karolina Dziugaite** [3]  **Ioannis Mitliagkas** [3]

## Abstract

Noise-based certified machine unlearning currently faces a hard ceiling: the noise magnitude required to certify unlearning typically destroys model utility, particularly for large-scale deletion requests. While leveraging public data is a standard technique in differential privacy to relax this tension, its role in unlearning remains unexplored. We address this gap by introducing **Asymmetric Langevin Unlearning (ALU)**, a framework that uses public data to mitigate privacy costs. We prove that public data injection suppresses the unlearning cost by a factor of $O(1/n_{\text{pub}}^2)$, guaranteeing a strict computational advantage over retraining. This establishes a new control mechanism: practitioners can mitigate the need for high noise—and the associated utility loss—by increasing the volume of public data. Crucially, we analyze the realistic setting of **distribution mismatch**, explicitly characterizing how shifts between public and private sources impact utility. We show that ALU enables "mass unlearning" of constant dataset fractions – a regime where standard symmetric methods become impractical – while maintaining high utility. Empirical evaluations using variational Rényi divergence and membership inference attacks confirm that ALU effectively thwarts privacy attacks while preserving utility under reasonable distribution shifts.

## 1. Introduction

The widespread adoption of machine learning across diverse applications has prompted regulatory responses aimed at protecting user privacy and data rights. Legislative frameworks such as the European Union's AI Act (Parliament & of the European Union, 2024) and Canada's Artificial Intelligence and Data Act (AIDA) (Parliament of Canada, 2022) establish fundamental principles including the "right to be forgotten," which mandates that individuals can request removal of their personal data from trained systems. While the most straightforward approach to these requests—retraining from scratch—provides perfect guarantees, it is computationally prohibitive for modern deep learning models. Consequently, the field has gravitated toward approximate methods, particularly the family of *noise-based injection and fine-tuning*, such as Langevin Unlearning (Chien et al., 2024a; Koloskova et al., 2025). These methods offer certifiable privacy guarantees but operate under a strict trade-off: the magnitude of noise required to certify the data erasure degrades the model's utility.

In this work, we address this limitation by exploring an idea established in Differential Privacy (DP) but unexplored in machine unlearning: the integration of *public data*. We operate under the realistic assumption that while sensitive user data must be unlearnable, there often exists a corpus of public data that is not subject to retraction requests. We propose **Asymmetric Langevin Unlearning (ALU)**, a framework that leverages this public data as a mechanism to improve the privacy-utility trade-off. To our knowledge, the only prior work exploring mixed-privacy unlearning is Golatkar et al. (2021), who introduced Mixed-Linear Forgetting for computer vision tasks. Their approach requires architectural modifications to achieve forgetting through network linearization, limiting its applicability. In contrast, ALU operates directly on standard training pipelines. Intuitively, public data acts as a stability anchor; it ensures that the weight distributions of the originally trained model and the retrained model remain naturally close. This proximity reduces the need for noise injection to bridge the gap between distributions, thereby preserving utility.

- We prove that injecting public data creates a more favorable initialization for the unlearning process, reducing the unlearning cost by a factor of $\mathcal{O}(1/n_{\text{pub}}^2)$ (Theorems 3.1 and 3.2). This structural advantage enables two capabilities:

---

[1]Université de Montréal [2]Mila, Quebec AI Institute [3]Google DeepMind. Correspondence to: Ahmed Mehdi Inane <mehdi-inane.ahmed@mila.quebec>.

*Proceedings of the 43rd International Conference on Machine Learning*, Seoul, South Korea. PMLR 306, 2026. Copyright 2026 by the author(s).

– **Mass Unlearning:** Unlike prior methods where noise requirements are independent of total dataset size, ALU allows for unlearning *constant fractions* of the private dataset (Corollary 3.1), ensuring robustness to large-scale deletion.

– **Computational Advantage:** We revisit the efficiency of unlearning compared to retraining from scratch. While standard Langevin Unlearning is known to be efficient asymptotically, we establish a stronger result: ALU maintains a strict computational advantage over retraining even in finite-step regimes.

• We depart from the idealized assumption of identical private-public distributions. We derive a new generalization bound Theorem 4.1 that explicitly quantifies the trade-off between the benefits of noise reduction, and the penalties of distribution mismatch between public and private sources.

• We introduce a rigorous evaluation methodology using variational estimation of the Rényi divergence to validate our bounds. Our experiments confirm that ALU successfully defends against membership inference attacks (U-LiRA) while preserving significantly higher utility than symmetric baselines.

## 2. Related work

### 2.1. Machine unlearning

Machine unlearning algorithms eliminate the influence of designated training data (the *forget set*) while balancing unlearning efficacy, model utility, and computational efficiency. Three canonical strategies illustrate the trade-offs: random re-initialization achieves perfect unlearning but destroys utility; retraining from scratch provides optimal guarantees but incurs prohibitive costs; no intervention preserves utility but achieves no unlearning. Unlearning paradigms diverge between **exact unlearning**, which matches the retraining baseline but limits expressivity or efficiency (Cao & Yang, 2015; Yan et al., 2022), and **approximate unlearning**, which provides certified approximations of retraining (Nguyen et al., 2020; Guo et al., 2023; Chien et al., 2024b; Koloskova et al., 2025).

Recent advancements in machine unlearning have explored many strategies to mitigate the computational burden of retraining. Some methods prioritize efficiency by exclusively modifying or isolating specific components of the model, such as the final layers or via parameter-efficient fine-tuning modules (Chowdhury et al., 2025). While effective in practice, these approaches either lack formal theoretical guarantees or require fundamental alterations to standard training pipelines and architectures, akin to SISA (Bourtoule et al.,

2020). Another line of work investigates unlearning in restricted settings, such as when the source data is entirely inaccessible. However, these solutions rely on deterministic algorithms that assume model convergence (Ahmed et al., 2025).

In contrast to these paradigms, our work operates within the randomized frameworks of approximate unlearning, namely $(\varepsilon, \delta)$-unlearning and Rényi unlearning, where schemes like noisy fine-tuning on the retain set have been extensively studied (Chien et al., 2024a;b). We investigate specifically the unexplored benefits of incorporating public data, in the general framework of Langevin Unlearning. We highlight that this is not an algorithmic contribution, but a characterization of how the unlearning utility trade-off is improved by the asymmetric nature of this setting.

### 2.2. Langevin Unlearning

A common approach to machine unlearning is to run a noisy projected gradient method starting from the trained weights, targeting a distribution close to retraining. Formally, at iteration $t$,

$$\theta_{t+1} = \Pi_\Theta[\theta_t - \eta\nabla_\theta\mathcal{L}(\theta_t) + \xi_t], \qquad (1)$$

where $\mathcal{L}$ is a surrogate loss (e.g., empirical loss on a retain set), $\eta$ is the step size, and $\xi_t$ is injected noise (often Gaussian) controlling distributional closeness.

Langevin Unlearning (LU) (Chien et al., 2024a) instantiates this scheme with $\mathcal{L} = \mathcal{L}_{\mathcal{D}_r}$, the loss on the retain set, and $\xi_t \sim \mathcal{N}(0, I_d)$. This reduces to projected noisy gradient descent (PNGD) (pseudocode in Appendix E).

LU provides certifiable approximate unlearning guarantees by minimizing the Rényi divergence between post-unlearning and post-retraining weight distributions (Chien et al., 2024a;b). However, these guarantees require that the *entire original training process* satisfies differential privacy (DP). This necessitates injecting substantial noise starting from the very first training iteration, which compromises the utility of the base model even before any unlearning request. In this work, we improve upon Chien et al. (2024a) by leveraging public data to mitigate this "privacy tax". We demonstrate that anchoring the training process with public data allows us to reduce the noise magnitude required during both learning and unlearning while satisfying the same guarantees. By assuming the initialization satisfies a log-Sobolev inequality—a mild condition satisfied by Gaussian initialization—we derive data-dependent bounds showing that public data acts as a stabilizer, effectively "subsidizing" the privacy cost of the private data. Concurrent approaches like (Koloskova et al., 2025) require only smoothness assumptions, but such data-agnostic bounds depend primarily on projection set geometry rather than exploiting the structural advantage of public data.

# 3. Asymmetric Langevin Unlearning

## 3.1. Preliminaries

**Motivation.** Our approach is motivated by a realistic data setting, well-established in the privacy machine learning literature (Alon et al., 2019; Amid et al., 2022; Ganesh et al., 2023; Lowy et al., 2024), that leverages public data to improve the privacy-utility trade-off. We introduce this asymmetric data model to Langevin Unlearning, which allows us to relax the restrictive Differential Privacy (DP) assumption over the entire dataset. By explicitly modeling this asymmetry, we can leverage public data to enhance the unlearning process to improve both efficacy and model performance without compromising privacy guarantees.

**Notation.** We consider probability distributions defined over a compact parameter space $\Theta$, where stochasticity arises from three sources: the weight initialization distribution $\pi_0$, the training data distribution $P_{\text{train}}$, and the inherent randomness of the optimization procedure. We denote by $\mathcal{P}(\Theta)$ the set of probability distributions supported on $\Theta$. We study the weight distributions $\pi_S^t$, where $S \in \{L, U, R\}$ identifies the training regime and $t$ denotes the iteration count. A key quantity in our analysis is the Rényi divergence of order $\alpha$ between distributions $P$ and $Q$, denoted $D_\alpha(P\|Q)$ (Definition 3.2). We use $P_{\text{pub}}$ and $P_{\text{priv}}$ to represent the distributions of public and private data, respectively.

**Problem Setting.** We consider empirical risk minimization over a dataset $D = D_{\text{pub}} \cup D_{\text{priv}}$ comprising two components: a public set $D_{\text{pub}}$ with $n_{\text{pub}}$ samples from a distribution $P_{\text{pub}}$, and a private set $D_{\text{priv}}$ with $n_{\text{priv}}$ samples from a distribution $P_{\text{priv}}$. The training loss is $\mathcal{L}_D(\theta) = \frac{1}{n_{\text{pub}}+n_{\text{priv}}} \sum_{x \in D} \ell(\theta, x)$. Only the private data is subject to unlearning requests, while public data remains permanently available. We employ $T$ PNGD iterations with projections onto $\Theta \subset \mathbb{R}^d$ (radius $R$) to obtain $\theta_T$. Since PNGD injects Gaussian noise at each step, it induces probability distributions over the parameter space. To ensure convergence and certifiable guarantees, we assume the initialization distribution satisfies a Log-Sobolev inequality (LSI):

**Definition 3.1.** *(Log-Sobolev inequality (Gross, 1975)) A probability measure $P \in \mathcal{P}(\mathbb{R}^d)$ satisfies a Log-Sobolev inequality with constant $C$ if*

$$\forall Q \in \mathcal{P}(\mathbb{R}^d), D_{KL}(Q\|P) \le \frac{C}{2} I(Q, P), \qquad (2)$$

*where $D_{KL}$ denotes the KL divergence and $I(Q, P) = E_Q \left[ \|\nabla \log \frac{q}{p}\|^2 \right]$ is the relative Fisher information.*

**Weight Distributions** We analyze three distributions induced by PNGD on $D = D_{\text{pub}} \cup D_{\text{priv}}$ given a request $D_{\text{forget}} \subseteq D_{\text{priv}}$ (Figure 1):

- *Learning distribution $\pi_L^T$*: results from $T$ iterations on $D$ with $\theta_0 \sim \pi_0$, representing the pre-unlearning model;

- *Unlearning distribution $\pi_U^K$*: results from $K$ fine-tuning iterations on $D \setminus D_{\text{forget}}$ initialized from $\theta \sim \pi_L^T$;

- *Retraining distribution $\pi_R^T$*: results from $T$ iterations on $D \setminus D_{\text{forget}}$ with $\theta_0 \sim \pi_0$, serving as the retraining baseline.

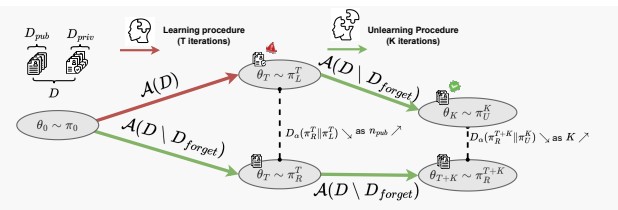

*Figure 1.* Training pipelines showing the relationship between learning, unlearning, and retraining with public data injection. The divergence $D_\alpha(\pi_R^T\|\pi_L^T)$ quantifies how public data helps maintain similarity between retraining and original learning distributions, facilitating subsequent unlearning.

Following Chien et al. (2024a), we measure unlearning quality via Rényi divergence.

**Definition 3.2.** *For probability measures $P, Q$ with $P \ll Q$, their Rényi divergence of order $\alpha \in (0, +\infty) \setminus \{1\}$ is*

$$D_\alpha(P\|Q) = \frac{1}{\alpha - 1} \log \mathbb{E}_Q \left[ \left( \frac{dP}{dQ} \right)^\alpha \right],$$

*where $\frac{dP}{dQ}$ is the Radon-Nikodym derivative. This generalizes KL divergence ($\alpha \to 1$), reverse-KL ($\alpha \to 0$), and connects to $\varepsilon$-differential privacy in the limit $\alpha \to \infty$ (Mironov, 2017).*

The effectiveness of unlearning is measured by $D_\alpha(\pi_U^K\|\pi_R^{T+K})$, while the presence of public data helps control $D_\alpha(\pi_R^T\|\pi_L^T)$, creating favorable conditions for the unlearning process.

## 3.2. Unlearning Performance

We now present theoretical guarantees for asymmetric Langevin unlearning that demonstrate how public data improves unlearning efficiency. Our analysis adapts the prior work of Chien et al. (2024a) by relaxing global differential privacy assumptions, and providing explicit characterization of how public and private data contributions differ in the unlearning bounds. The following result explains how public data reduces reliance on differential privacy constraints:

**Theorem 3.1** (The role of public data in shrinking the learning / retraining mismatch.)**.** *Suppose that the loss is $L$-smooth and $M$-Lipschitz, and that the initialization distribution satsifies a $C_0$-log Sobolev inequality. Moreover,*

*suppose that the PNGD updates project onto a compact set $\Theta$ of radius $R$.*

*Then at learning iteration $T$, we have the following upper bound on the Rényi divergence between the retraining $\pi_R^T$ and learning $\pi_L^T$ distributions:*

$$\frac{D_\alpha(\pi_R^T \| \pi_L^T)}{\alpha} \leq \frac{2M^2\eta^2 n_{\text{forget}}^2}{(n_{\text{pub}} + n_{\text{priv}})^2 \sigma^2} \sum_{t=1}^{T-1} \prod_{t'=t}^{T-1} h(t', \eta, \sigma),$$

*where $h(t', \eta, \sigma) = \left(1 + \frac{\eta\sigma^2}{C_{t',1}}\right)^{-1}$, and $0 < C_{t',1} \leq (1 + \eta L)^{2K} C_0 + 2\eta\sigma^2 \frac{(1+\eta L)^{2K}-1}{(1+\eta L)^2 - 1}$ are log Sobolev constants of the distributions of the intermediate PNGD updates. Using the support's radius allows to loosely upper bound those constants (Chien et al., 2024a): $C_{t',1} \leq 6 e^{\frac{4\tau}{\eta\sigma^2}} (4\tau^2 + \eta\sigma^2)$ with $\tau = R + \eta M$.*

*Proof sketch. The proof follows the analytical framework of Chien et al. (2024a, Theorem 3.3), adapted to leverage the presence of public data in the training set. By distinguishing between public and private data contributions in the gradient updates, we reduce the privacy erosion (Chourasia et al., 2021) of each PNGD update. Full proof details are presented in Appendix B.1.*

This bound reveals that we can fix noise magnitude $\sigma$ to be arbitrarily small to preserve performance while controlling the divergence through public data volume. When $n_{\text{pub}} \gg n_{\text{forget}}$, the learning and retraining distributions remain close regardless of noise level, providing favorable initial conditions for unlearning (Figure 3b). Specifically, the public data reduces the squared sensitivity of the gradient updates by a factor of $(n_{pub} + n_{priv})^2$.

This dependency allows us to characterize a regime where asymmetric unlearning provides meaningful guarantees while symmetric methods fail: the unlearning of a constant fraction of the private dataset.

**Corollary 3.1** (LU noise vs ALU noise). *Consider the setting where the forget set size constitutes a constant fraction of the private data (i.e., $n_{\text{forget}} = cn_{\text{priv}}$ for some $c \in (0, 1]$). Let $\sigma_{\text{sym}}^2$ and $\sigma_{\text{asym}}^2$ be the noise variances required to bound the Rényi divergence between the learning and retraining distributions by a fixed $\varepsilon$ in the symmetric and asymmetric settings, respectively. These variances satisfy the following lower bounds:*

$$\sigma_{\text{sym}}^2 \geq \frac{2M^2\eta^2 c^2}{\varepsilon} \sum_{t=1}^{T-1} \prod_{t'=t}^{T-1} h(t', \eta, \sigma),$$

$$\sigma_{\text{asym}}^2 \geq \frac{2M^2\eta^2 c^2}{\varepsilon} \left(\sum_{t=1}^{T-1} \prod_{t'=t}^{T-1} h(t', \eta, \sigma)\right) \left(\frac{n_{\text{priv}}}{n_{\text{priv}} + n_{\text{pub}}}\right)^2.$$

*Proof sketch. The proof follows immediately from Theorem 3.1, by considering that $\frac{n_{\text{forget}}}{n_{\text{public}}+n_{\text{priv}}} = c \frac{n_{\text{priv}}}{n_{\text{public}}+n_{\text{priv}}}$.*

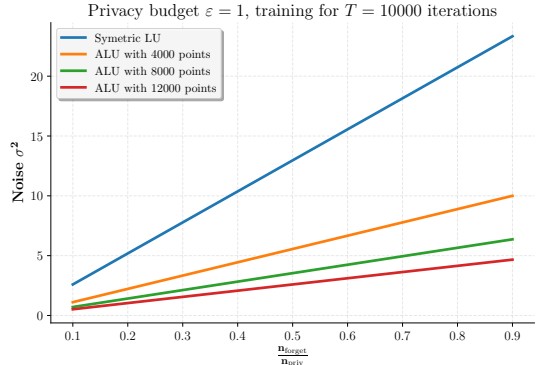

Privacy budget $\varepsilon = 1$, training for $T = 10000$ iterations

*Figure 2.* **Required noise magnitude $\sigma$ to bound $D_\alpha(\pi_R^T \| \pi_L^T)$ as a function of the forget fraction $c = n_{\text{forget}}/n_{\text{priv}}$.** Values are computed assuming a strongly convex loss (Chien et al., 2024a), for a *binary* classification task. Details are deferred to Appendix B.3.

**Remark.** A limitation of the standard symmetric setting ($n_{\text{pub}} = 0$) is that the required noise variance $\sigma_{\text{sym}}^2$ has **no dependency on the total number of training points** when unlearning a constant fraction $c$. This implies that increasing the dataset size does not mitigate the unlearning cost, rendering the bound vacuous for large cohorts where the required high noise magnitude destroys utility.

In contrast, our framework introduces an explicit dependency on the dataset composition. The required noise scales with the ratio $\frac{n_{\text{priv}}}{n_{\text{priv}}+n_{\text{pub}}}$, meaning the sensitivity of the distribution effectively decays as public data is added. This allows the learning distribution to remain arbitrarily close to the retraining distribution—even for large forget sets—provided sufficient public data is available. We illustrate this advantage in Figure 2.

**Theorem 3.2** (Convergence guarantee of Langevin unlearning (Chien et al., 2024a, Theorem 3.2)). *Suppose that the loss is $L$-smooth and $M$-Lipschitz, and that the learning distribution of weights at time $T$ satisfies a $C$ log-Sobolev inequality. Then, the Rényi divergence between $\pi_U^K$ (the unlearning distribution after $K$ iterations) and the retraining distribution after $T + K$ iterations is upper bounded by*

$$D_\alpha(\pi_R^{T+K} \| \pi_U^K) \leq D_\alpha(\pi_L^T \| \pi_R^T)$$
$$\times \min\left(g_{\alpha,\eta,L}(k, \sigma), \exp\left(-\frac{2K\sigma^2\eta}{\alpha\tilde{C}}\right)\right),$$

*where $g_{\alpha,\eta,L}(k, \sigma) = \prod_{k=1}^{K}\left(1 + \frac{2\eta\sigma^2}{(1+\eta L)^2 C_{U,k}}\right)^{-1/\alpha}$, and $0 < C_k \leq (1 + \eta L)^{2K}C + 2\eta\sigma^2\frac{(1+\eta L)^{2K}-1}{(1+\eta L)^2-1}$, and $\tilde{C} \leq 6\left(4\tau^2 + 2\eta\sigma^2\right)\exp\left(\frac{4\tau^2}{2\eta\sigma^2}\right)$.*

*Moreover, if the loss function is $m$-strongly convex and the initial log-Sobolev constant satisfies $C > \frac{\sigma^2}{m}$, we get the following exponential decay of the Rényi divergence with*

*respect to the unlearning iteration:*

$$D_\alpha(\pi_R^{T+K}\|\pi_U^K) \leq D_\alpha(\pi_L^T\|\pi_R^T)\exp\left(-\frac{2K\sigma^2\eta}{C\alpha}\right).$$

This theorem establishes the convergence guarantee for Langevin unlearning by showing that the Rényi divergence between the unlearning and retraining distributions decreases exponentially with unlearning iterations $K$, with the convergence rate controlled by the initial divergence $D_\alpha(\pi_R^{T+K}\|\pi_U^K)$. When combined with Theorem 3.1, this reveals the mechanism by which public data improves unlearning: the quadratic reduction in initial divergence from public data injection translates directly into tighter convergence bounds.

**Remark (On the tractability of Log-Sobolev constants)** While Theorems 3.1 and 3.2 provide explicit convergence bounds, we acknowledge that the Log-Sobolev constants (LSI) $(C_t, \tilde{C})$ are difficult to estimate for deep neural networks and may scale poorly with dimension. However, our core contribution—the role of public data—operates independently of these absolute constants. Specifically, the quadratic reduction in gradient sensitivity (Theorem 3.1) acts as a multiplicative factor that improves the divergence bound relative to the baseline, regardless of the specific value of the LSI constant. Thus, even if the absolute convergence rate is loose due to a large $C$, the relative advantage of injecting public data remains theoretically guaranteed.

**Computational advantage** Assuming the loss is $m$-strongly convex, the computational advantage of ALU over retraining is determined by the training budget $T$, the privacy requirement $\varepsilon$ and the fraction of private data to unlearn $n_{\text{forget}}/n_{\text{priv}}$. When $T$ is small, unlearning is more expensive than retraining in traditional settings $(n_{\text{public}} = 0)$. ALU overcomes this via the public data ratio, remaining efficient $(K < T)$ only if $n_{\text{public}}/n_{\text{priv}} \geq \mathcal{C} \cdot T^\beta \cdot \varepsilon^{-1/2} \cdot (n_{\text{forget}}/n_{\text{priv}}) - 1$, where $\beta = \sigma^2\eta/C\alpha$. Here, public data acts as a buffer that facilitates distribution alignment without consuming the privacy budget. Conversely, in the large $T$ regime, the cost of unlearning scales as $\mathcal{O}(\log(n_{\text{total}}))$ (Chien et al., 2024a). In this state of convergence, the addition of any data reduces the initial divergence, ensuring unlearning is strictly more efficient than retraining. See Appendix D and Proposition D.1 for a derivation of a sufficient condition characterizing the computational advantage of unlearning over retraining.

# 4. Performance Without Noise: The Role of Distribution Alignment

LU faces a fundamental dilemma: increasing noise improves unlearning guarantees but degrades model performance.

Our asymmetric approach breaks this trade-off by leveraging public data abundance rather than noise amplification. However, the effectiveness of this strategy depends on the relationship between public and private data distributions.

We now analyze when public data injection preserves performance, and when it introduces new challenges. Our results reveal that performance preservation is not automatic – it depends on the distributional alignment between public and private data. When these distributions are similar, public data acts as a performance stabilizer, allowing effective unlearning without quality degradation. Conversely, when distributions differ significantly, performance impacts emerge, though they remain more controlled than noise-based approaches.

We evaluate post-unlearning performance on the private data distribution *only*, reflecting realistic deployment scenarios where the primary concern is maintaining model quality on the sensitive data that remains after unlearning. Performance analysis on the full mixture of public and private distributions is provided in Appendix C for completeness.

**Theorem 4.1** (Generalization Bound under Distribution Mismatch). *Assuming the data generating distributions share the same support, that the weight space $\Theta$ is compact and that the loss is $M$-Lipschitz wrt $\theta$, we have the following upper bound on the generalization error on the private data after performing $K$ iterations of unlearning, and initializing a weight $\theta_0$ from $\pi_L^T$:*

$$\mathbb{E}_{\pi_U^K}[\mathcal{L}_{P_{priv}}] \leq \underbrace{e^{\frac{n_{pub}}{n_{pub}+n_{retain}}D_\infty(P_{priv}\|P_{pub})}}_{\text{distribution mismatch penalty}}\mathbb{E}_{\pi_R^{T+K}}[\mathcal{L}_{P_{train}}]$$
$$+ M \cdot diam(\Theta) \cdot \underbrace{\sqrt{\frac{1}{2}D_\alpha(\pi_R^{T+K}\|\pi_U^K)}}_{\text{unlearning approximation error}}$$

*where $\mathcal{L}_P(\theta) := \mathbb{E}_{x\sim P}[\ell(\theta, x)]$ denotes the risk over distribution $P$; $P_{\text{train}}$ is the mixture of $P_{\text{pub}}$ and $P_{\text{priv}}$ used during training; and $D_\infty(P\|Q) = \log\left(\text{ess sup}\frac{p}{q}\right)$ is the infinite Rényi divergence representing the worst-case regret (Erven & Harremoës, 2014).*

*Proof sketch. The proof uses the Kantorovitch-Rubinstein duality (Theorem C.1) to bound the performance gap by the dual of the Wasserstein distance between $\pi_U^K$ and $\pi_L^{T+K}$, then relates this to Rényi divergence via standard inequalities leveraging the compactness of the weight space $\Theta$. For private data evaluation, importance weighting introduces a mismatch penalty controlled by the worst case regret, $D_\infty(P_{\text{priv}}\|P_{\text{pub}})$, weighted by the public data fraction. See Appendix C for full proof details.*

When $n_{\text{pub}} \to \infty$:

1. **Aligned distributions** $(D_\infty(P_{\text{priv}}\|P_{\text{pub}}) \approx 0)$: The

distribution mismatch penalty vanishes, and the unlearned model's performance on unseen private data is guaranteed to be at least as good as the retrained model's performance on the training mixture. This represents the ideal scenario where public data injection preserves performance.

2. **Misaligned distributions** $(D_\infty(P_{\text{priv}} \| P_{\text{pub}}) \gg 0)$: The exponential penalty term dominates, causing the upper bound to become vacuous. While this confirms that performance degradation will occur, the bound's looseness prevents us from quantifying the actual extent of this degradation. The true performance impact may be better than this worst-case guarantee suggests.

# 5. Experiments

Our theoretical analysis provides upper bounds on the Rényi divergence $D_\alpha(\pi_R^{T+K} \| \pi_U^K)$ that governs unlearning performance. However, these bounds involve iteration-dependent log-Sobolev constants that are difficult to estimate in practice, making it unclear how tight our theoretical guarantees actually are. To gain empirical insight into the behavior of this divergence, we estimate its value using samples from the weight distributions. To our knowledge, this is the first attempt to evaluate unlearning performance through direct estimation of the Rényi divergence between the parameter distributions. Building on Birrell et al. (2021; 2023), we leverage the variational representation of the Rényi divergence for numerical estimation.

**Theorem 5.1.** *(Convex conjugate variational approximation of the Rényi divergence (Birrell et al., 2023)) Let P,Q two probability distributions supported on $\Omega$, such that $P \ll Q$, and let $\mathcal{M}_b$ be the space of bounded measurable functions on $\Omega$. Then, $\forall \alpha \in (0, +\infty) \setminus \{1\}$,*

$$\frac{D_\alpha(P \| Q)}{\alpha} = \sup_{g \in \mathcal{M}_b(\Omega), g < 0} \int g \, dQ$$
$$+ \frac{1}{\alpha - 1} \int |g|^{\frac{\alpha-1}{\alpha}} dP + \alpha^{-1} (\log \alpha + 1).$$

This variational representation of Rényi divergence allows us to obtain estimates of $D_\alpha(\pi_R^{T+K} \| \pi_U^K)$ using trained models as samples. We emphasize that **this is not intended as a practical evaluation methodology for machine unlearning**, as it requires training numerous models to obtain sufficient samples for reliable statistical estimation. Standard approaches like membership inference attacks (MIAs) (Shokri et al., 2017; Carlini et al., 2021; Hayes et al., 2024) remain more suitable for practical evaluation. Our goal is purely investigative: to understand how the Rényi divergence behaves empirically and assess whether our theoretical bounds, despite containing intractable constants.

We present our findings in two parts: Sections 5.1 and 5.2 investigate the behaviour of the upper bounds provided respectively in Theorem 3.2 and Theorem 4.1, while Section 5.3 provides standard membership inference attack and utility evaluations to contextualize our approach within existing unlearning assessment practices.

## 5.1. Evaluating the Rényi Divergence

**Experimental Setup.** We evaluate our approach on a multi-class image classification task using two domains from the DomainNet dataset (Peng et al., 2019): Quickdraw (sketches) and Clipart (stylized images), each containing 24 classes. We select these visually distinct domains to investigate how public-private data alignment affects unlearning and utility (Figure 5).

The experimental configuration treats Clipart images as private data (subject to unlearning) and Quickdraw images as public data (permanently retained). For a training set of size $n = n_{\text{pub}} + n_{\text{priv}}$, we train models using cross-entropy loss and PNGD updates. To obtain samples from the weight distributions $\pi_U^K$ and $\pi_R^T$, we train $N$ models in parallel: one set undergoes unlearning (fine-tuning on the retain set after initial training), while another set trains from scratch on the retain set only. This procedure yields $N$ weight samples from each distribution. We approximate the variational Rényi representation (Equation (Lemma A.2)) using neural network discriminators to parameterize the function space $\mathcal{M}_b(\Omega)$. This approach follows established practices in divergence estimation (Birrell et al., 2021; 2023; Belghazi et al., 2021) (pseudo-code in Appendix G.1).Complete details on discriminator architecture and training procedures are provided in Appendix G.

**Results.** Figure 3a presents our Rényi estimation results, demonstrating the effectiveness of public data injection for improving unlearning efficiency. The experiments are conducted using $N = 30000$ models for each distribution and averaged across 5 discriminator trainings with spectral normalization. The PNGD noise scale is $\sigma = 0.01$ and $\alpha = 2$. The results show that increasing public data volume reduces $D_\alpha(\pi_R^{T+K} \| \pi_U^K)$, with the divergence decreasing both as a function of unlearning iterations and public data proportion. To understand the mechanism driving these improvements, we conduct an ablation study examining the initial conditions after a *single* unlearning iteration. Figure 3b isolates the effect of public data on the starting distributions by measuring $D_\alpha(\pi_R^{T+1} \| \pi_U^1)$ as a function of public data volume. Rather than directly improving the unlearning procedure itself, public data creates more favorable initial conditions by ensuring the learning and retraining weight distributions begin in closer proximity. This mechanistic understanding validates our theoretical framework: public data primarily controls the initial gap between distributions (Theorem 3.1),

which then propagates through the unlearning iterations to produce the final performance gains.

## 5.2. Distribution Alignment and the Unlearning-Utility Trade-off

Theorem 4.1 characterizes a trade-off caused by public data injection: as we increase public data volume, the *unlearning approximation error* decreases, yet the *distribution mismatch penalty* simultaneously grows. The balance between these competing terms determines whether public data injection preserves or degrades model performance. To empirically investigate this trade-off, we conduct experiments across two distinct distributional regimes: one where the public and private domains exhibit moderate visual alignment, and another where they are substantially misaligned.

We fix $K = 5$ unlearning iterations and evaluate performance using the DomainNet dataset across two domain pairs. The **aligned regime** pairs Quickdraw (public) and Clipart (private), which despite visual stylistic differences share semantic structure. The **misaligned regime** pairs Infograph (public) and Real (private), which exhibit greater distributional divergence. We measure model performance via loss on the private data distribution $P_{\text{priv}}$ after unlearning, comparing against the retraining baseline on the training mixture. Results are summarized in Table 1.

*Table 1.* Comparing Unlearning vs Retraining Performance Across Distribution Alignments ($K = 5$, Private: 20000 points, Forget Set: 10000 points)

| Public Domain | Private Domain | Public Points | Unlearn Loss | Retrain Loss | Rel. Diff (%) |
|---|---|---|---|---|---|
| Quickdraw | Clipart | 10000 | 3.102 | 2.976 | 4.23 |
| Quickdraw | Clipart | 30000 | 3.102 | 2.965 | 4.62 |
| Quickdraw | Clipart | 40000 | 3.099 | 2.989 | 3.68 |
| Infograph | Real | 10000 | 2.233 | 2.495 | 10.53 |
| Infograph | Real | 30000 | 2.238 | 2.496 | 10.34 |
| Infograph | Real | 40000 | 2.233 | 2.504 | 10.81 |

The results reveal a contrast between the two regimes. In the **aligned setting**, the relative performance gap remains modest (3.68–4.62%) across varying public data volumes, suggesting that the mismatch penalty remains manageable and the approximation error reduction dominates. In contrast, the **misaligned setting** exhibits a persistent performance gap (10.34–10.81%), with minimal sensitivity to public data volume. This indicates that when distributional divergence is large, increasing public data fails to overcome the mismatch penalty, rendering the approximation error reduction insufficient to improve generalization.

## 5.3. Practical Evaluation of LU in the Asymmetric Setting

We now adopt standard evaluation methodology from the unlearning literature (Hayes et al., 2024), highlighting the benefit of public data injection. We provide an overview here and defer details to Appendix H.

**Evaluation Method.** This evaluation is based on the U-LiRA membership inference attack for unlearning (Hayes et al., 2024; Carlini et al., 2021). Given a training set, forget set, and specified learning and unlearning algorithms, the adversary's goal is to infer whether a model's weights $\theta$ were drawn from the unlearning distribution $\pi_U^K$ or the retraining distribution $\pi_R^{T+K}$. Intuitively, lower attack accuracy indicates that the unlearning and retraining distributions are harder to distinguish, i.e., better unlearning.

In its most basic form, U-LiRA can be formalized via Bayes' rule under a uniform prior on whether the forget set was included during training. Letting $P(\theta \mid \cdot)$ denote the likelihood of observing model parameters $\theta$ under a given distribution, and $P(\cdot \mid \theta)$ as the posterior probability that $\theta$ was drawn from that distribution, we have

$$P(\pi_U^K \mid \theta) = \frac{P(\theta \mid \pi_U^K)}{P(\theta \mid \pi_U^K) + P(\theta \mid \pi_R^{T+K})}.$$

By selecting a one-dimensional representation of the models $f : \Theta \to \mathbb{R}$ and assuming that the induced distributions $f_\sharp \pi_U^K$ and $f_\sharp \pi_R^{T+K}$ are Gaussian, we can estimate the likelihood terms $P(\theta \mid \cdot)$ from a tractable number of model samples.

**Experimental Setup.** For the sake of completeness, we focus this next set of experiments on a sentiment analysis task on the IMDB dataset of movie reviews (Maas et al., 2011). This is a binary classification task, where an LSTM (Hochreiter & Schmidhuber, 1997) learns to recognize if a review is either negative or positive. We use the Amazon reviews dataset from Zhang et al. (2015) as the public data source. We use a forget set of 100 uniformly sampled examples from the IMDB dataset. For both experiments, i.e., with and without public data injection, we generate $N = 50$ models to estimate each likelihood density, and report the empirical distribution of probabilities assigned to the right origin distribution by U-LiRA (confidence scores) for 50 models test (25 from $\pi_U^K$, and 25 from $\pi_R^{T+K}$, where $T = 50$ and $K = 1 \to 15$). Figure 4 highlights that without public data injection, U-LiRA is able to identify a large proportion of models confidently and correctly, even after a number of unlearning steps. This observed discriminative power is heavily impacted by public data injection. We can also observe that modes of the confidence scores generally decrease with the number of unlearning steps, highlighting the unlearning effectiveness of LU.

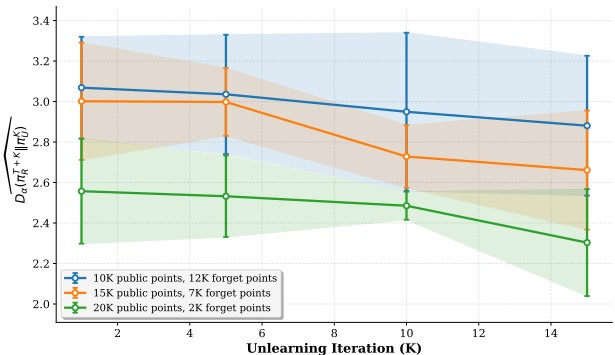
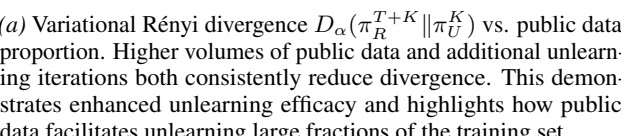

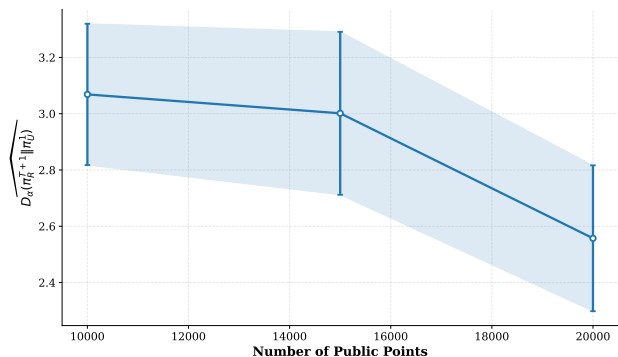

*(a)* Variational Rényi divergence $D_\alpha(\pi_R^{T+K} \| \pi_U^K)$ vs. public data proportion. Higher volumes of public data and additional unlearning iterations both consistently reduce divergence. This demonstrates enhanced unlearning efficacy and highlights how public data facilitates unlearning large fractions of the training set.

*(b)* Ablation study: initial distribution alignment ($K = 1$). The divergence $D_\alpha(\pi_R^{T+1} \| \pi_U^1)$ decreases as the public data volume increases, validating the more favorable initialization provided by public data injection.

*Figure 3.* Rényi divergence estimation across varying public data (Clipart) volumes.

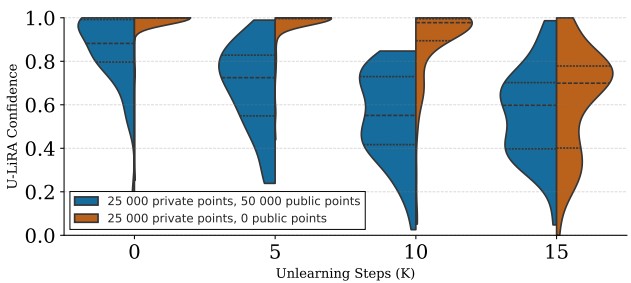

*Figure 4.* U-LiRA confidence scores after K unlearning iterations as violin plots with quartiles.

Now that we've observed the effect of public data injection on the unlearning effectiveness of LU, we change our focus towards its impact on model utility. To this end, we report in Table 2 the average model accuracies over the 75 models we trained for each model distribution, on a test set of $10,000$ unseen samples from the IMDB dataset. As the Amazon reviews dataset appears to be a good auxiliary public data source for the IMDB review classification problem (close data distributions), we also include an experiment in which a uniformly sampled $40\%$ of its labels are flipped, thus increasing distribution mismatch between public and private sources.

*Table 2.* Comparing unlearning against retraining test accuracies with flipped labels (IMDB, 25000 private points, $K = 15$ unlearning iterations)

| Public Dataset (Points) | Flipped Labels | Unlearned Acc. (%) | Retrained Acc. (%) |
|---|---|---|---|
| None (0) | 0% | 82.59 | 82.54 |
| Amazon Reviews (50 000) | 0% | 81.42 | 82.15 |
| Amazon Reviews (50 000) | 40% | 80.40 | 80.80 |

From Table 2, we can observe that model accuracy does

decrease from the injection of public data. However, this drop in accuracy is rather negligible compared to the extent to which public data injection improves the unlearning effectiveness of LU, which is highlighted by Fig. 4. As expected, the drop in accuracy is proportionately much lower when the quality of auxiliary public data is high ($1.17\%$ for unlearned and $0.39\%$ for retrained) than when it is low ($2.19\%$ for unlearned, an $\approx 1.87$ times increase, and $1.74\%$ for retrained, an $\approx 4.46$ times increase).

## 6. Future Work

Our framework opens several compelling avenues for research into the interplay of public data and unlearning. Algorithmic extensions include studying ALU in pretraining/fine-tuning regimes and developing adaptive algorithms that leverage domain adaptation to optimally balance distribution alignment with unlearning efficiency. We also envision a constrained optimization approach where the unlearning objective is regularized to keep the weight distribution close to a public-only reference.

Theoretically, addressing the intractable log-Sobolev constants in current Langevin analysis remains a priority. Transitioning to alternative isoperimetric assumptions (Chewi et al., 2021; Mousavi-Hosseini et al., 2023; Altschuler & Chewi, 2024) or adopting weaker divergence measures could yield more tractable bounds than those provided by Rényi divergence. Finally, extending analysis from weight distributions to output distributions would facilitate black-box evaluation.

Finally, as highlighted by Tramèr et al. (2024), large-scale datasets scraped from the web are not inherently privacy-neutral and may contain sensitive or copyrighted information. Applications should ensure that public datasets used

for initialization do not introduce secondary leakage, ensuring that the efficiency gains of ALU do not come at the cost of unintended exposure from the public source.

## 7. Conclusion

We have studied Langevin unlearning under the assumption of asymmetric data sources, where datasets contain both private and public data. Our theoretical analysis demonstrates that this framework fundamentally improves the unlearning-utility trade-off by enabling control over unlearning guarantees through data supplementation rather than noise amplification. The framework provides fine-grained analysis of how distributional alignment between public and private data affects this trade-off: when distributions are well-aligned, public data injection preserves utility while maintaining unlearning guarantees, while misaligned distributions introduce controlled performance penalties that remain more manageable than traditional noise-based approaches.

## Acknowledgments

We would like to thank Jeremiah Birrell, Eleni Triantafillou and Ryan d'Orazio for the helpful discussions. This research was enabled in part by compute resources, software and technical help provided by Mila (mila.quebec). Ahmed Mehdi Inane's and Vincent Quirion's research is supported by Ioannis Mitliagkas' CIFAR AI chair.

## Impact statement

This paper presents work whose goal is to advance the field of Machine Learning. There are many potential societal consequences of our work, none which we feel must be specifically highlighted here.

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

# A. Unlearning performance bounds

## A.1. Proof of Theorem 3.2

**Theorem.** *(Chien et al., 2024a) Suppose that the loss is $L$-smooth and $M$-Lipschitz, and that the learning distribution of weights at time $T$ satisfies a $C$ log-Sobolev inequality. Then, the Rényi divergence between $\pi_U^K$ (the unlearning distribution after $K$ iterations) and the retraining distribution after $T + K$ iterations is upper bounded by:*

$$D_\alpha(\pi_R^{T+K}\|\pi_U^K) \leq D_\alpha(\pi_L^T\|\pi_R^T)\exp\left(-\frac{1}{\alpha}\sum_{k=0}^{K-1} R_k\right)$$

*where $R_k > 0$ depend on the problem setting (Chien et al., 2024a). Moreover, if the loss function is $m$-strongly convex and the initial log-Sobolev constant satisfies $C > \frac{\sigma^2}{m}$, we get the following exponential decay of the Rényi divergence with respect to the unlearning iteration:*

$$D_\alpha(\pi_R^{T+K}\|\pi_U^K) \leq D_\alpha(\pi_L^T\|\pi_R^T)\exp\left(-\frac{2K\sigma^2\eta}{C\alpha}\right)$$

We provide the proof of (Chien et al., 2024a), Theorem 3.2, slightly modified to our setting. Specifically, we relax the assumption that the learning and retraining processes have converged to their stationary distribution (infinite training). In order to prove this theorem, we will use the following lemmas:

**Lemma A.1** (Characterizing the log-Sobolev constants of the PNGD updates (Chewi, 2023)). *Consider the PNGD update:*

$$\theta^{k+1} = \Pi_\Theta\left[\theta_k - \eta\nabla\mathcal{L}_D(\theta^k) + \sqrt{2\eta\sigma^2}W_k\right], \theta^0 \sim \pi$$

*where $\pi$ satisfies a $C$-Log Sobolev inequality. Then, we have the following:*

- *If $\mathcal{L}$ is $L$-smooth, then for the gradient update $h(\theta) = \theta - \nabla_\theta\mathcal{L}(\theta)$, we have that the distribution of $h_\sharp\pi$ satisfies a $(1 + \eta L)^2 \times C$ log-Sobolev inequality. Moreover, if $\mathcal{L}$ is $m$-strongly convex and $\eta < \frac{1}{L}$, then $h_\sharp\pi$ satisfies a $(1 - \eta m)^2 \times C$ log Sobolev inequality (Altschuler & Talwar, 2022).*

- *$\pi * \mathcal{N}(0, \sigma^2 I_d)$ satisfies a a $C + \sigma^2$ log-Sobolev inequality*

- *$\Pi_{\Theta\sharp}\pi$ satisfies a $C$ log-Sobolev inequality*

*By composing the aforementioned statements, we get that $\pi_1$ satisfies a $(1 + \eta L)^2 \times C + 2\eta\sigma^2$-log Sobolev inequality. Moreover, if $\mathcal{L}$ is $m$-strongly convex and $\eta < \frac{1}{L}$, we have that $\pi_1$ satisfies a $(1 - \eta m)^2 \times C + 2\eta\sigma^2$*

**Lemma A.2** (Data Processing inequality for the Rényi divergence (Erven & Harremoës, 2014)). *For any $\alpha \geq 1$, any function $h : \mathbb{R}^d \to \mathbb{R}^d$ and distributions $P, Q$ supported on $\mathbb{R}^d$, we have:*

$$D_\alpha(h_\sharp P\|h_\sharp Q) \leq D_\alpha(P\|Q)$$

*with equality if $h$ is bijective*

**Lemma A.3** ((Vempala & Wibisono, 2019; Chien et al., 2024a) characterizing the Rényi divergence between two distributions convoluted with Gaussians). *Let $P_t = P * \mathcal{N}(0, 2t\sigma^2 I_d)$ and $Q_t = Q * \mathcal{N}(0, 2t\sigma^2 I_d)$. Then, $\forall \alpha > 0$:*

$$\frac{\partial D_\alpha(P_t\|Q_t)}{\partial t} = -\alpha\sigma^2\frac{G_\alpha(P_t\|Q_t)}{F_\alpha(P_t\|Q_t)}$$

*with $G_\alpha(P\|Q) = \mathbb{E}_Q\left[\left(\frac{p}{q}\right)^\alpha\|\nabla\log\frac{p}{q}\|^2\right]$ denoting the relative Rényi information and $F_\alpha(P\|Q) = \mathbb{E}_Q\left[\left(\frac{p}{q}\right)^\alpha\right] = \exp((\alpha - 1)D_\alpha(P\|Q))$*

**Lemma A.4.** *Lower bound of the G-F ratio (Vempala & Wibisono, 2019) If $Q \in \mathcal{P}(\Theta)$ satisfies a $C$ log Sobolev inequality, then $\forall P \in \mathcal{P}(\Theta)$:*

$$\frac{G_\alpha(P\|Q)}{F_\alpha(P\|Q)} \geq \frac{2D_\alpha(P\|Q)}{\alpha^2 C}$$

**Lemma A.5.** *Grönwall's inequality (Gronwall, 1919) Let* $\mathbf{I} = [a, b]$ *denote an interval on the real line. Let* $\beta$ *and* $u$ *be real-valued continuous functions defined on* $\mathbf{I}$. *If* $u$ *is differentiable in the interior of* $\mathbf{I}$ *and satisfies for all* $t$ *in the interior of* $\mathbf{I}$:

$$\frac{\partial u(t)}{dt} \leq \beta(t)u(t)$$

*then we have:*

$$u(t) \leq u(a) \exp\left(\int_a^t \beta(s)ds\right)$$

*for all* $t \in I$

**Lemma A.6.** *Universal upper bound on the log Sobolev constant for measures with compact support (Chen et al., 2021) Let* $P$ *a probability measure supported on a compact set with radius* $R$. *Then, for each* $\sigma > 0$, $P * \mathcal{N}(0, \sigma I_d)$ *satisfy a log Sobolev inequality with constant upper bounded by* $6(4R^2 + \sigma) \exp\left(\frac{4R^2}{\sigma}\right)$

*Proof.* Using these results, we have:

$$D_\alpha(h_\sharp \pi_R^{T+K} \| h_\sharp \pi_U^K) \leq D_\alpha(\pi_R^{T+K} \| \pi_U^K) \qquad (\text{ Lemma A.2})$$

**The PNGD updates preserve the log-Sobolev inequality for the resulting distributions:** let $\pi_U^{K,1,t} = h_\sharp \pi_U^K * \mathcal{N}(0, 2t\sigma^2 I_d)$ and $\pi_R^{T+K,1,t} = h_\sharp \pi_U^K * \mathcal{N}(0, 2t\sigma^2 I_d)$. Since $\pi_L^T$ and $\pi_R^T$ satisfy a log-Sobolev inequality (initialization distributions) and the loss function is $L$-smooth, then by Lemma A.1 the distributions $\pi_U^K, \pi_L^{T+K}$ satisfy respectively $C_{U,K}, C_{L,T+K}$ log Sobolev inequalities. Using Lemma A.1 on the distributions $\pi_U^{K,1,t}, \pi_R^{T+K,1,t}$ yields that they respectively satisfy $(1 + \eta L)^2 C_{U,K} + 2\eta\sigma^2$ and $(1 + \eta L)^2 C_{L,T+K} + 2\eta\sigma^2$ log Sobolev inequalities for all $t \in [0, \eta]$.

**Upper bounding the distributions convolved with Gaussian distributions:** Using Lemma A.3, we have that, $\forall \alpha > 0$:

$$\frac{\partial D_\alpha(\pi_R^{T+K,1,t} \| \pi_U^{K,1,t})}{\partial t} = -\alpha\sigma^2 \frac{G_\alpha(\pi_R^{T+K,1,t} \| \pi_U^{K,1,t})}{F_\alpha(\pi_R^{T+K,1,t} \| \pi_U^{K,1,t})}$$

and since $\pi_U^{K,1,t}$ satisfies a $C_{U,K,t} = (1 + \eta L)^2 C_{U,K} + 2t\sigma^2$ log-Sobolev inequality, we can use Lemma A.4 to upper bound the derivative of the Rényi divergence with respect to $t \in [0, \eta]$:

$$\frac{\partial D_\alpha(\pi_R^{T+K,1,t} \| \pi_U^{K,1,t})}{\partial t} \leq -\frac{2\sigma^2}{\alpha C_{U,K,t}} D_\alpha(\pi_R^{T+K,1,t} \| \pi_U^{K,1,t})$$

Thus, by Grönwall's inequality (Lemma A.5), we have $\forall t \in [0, \eta]$:

$$\begin{aligned}
D_\alpha(\pi_R^{T+K,1,t} \| \pi_U^{K,1,t}) &\leq D_\alpha(h_\sharp \pi_R^{T+K} \| h_\sharp \pi_U^K) \exp\left(\int_0^t -\frac{2\sigma^2}{\alpha C_{U,K,s}} ds\right) \\
&\leq D_\alpha(h_\sharp \pi_R^{T+K} \| h_\sharp \pi_U^K) \exp\left(\int_0^t -\frac{2\sigma^2}{\alpha\left((1 + \eta L)^2 C_{U,K} + 2s\sigma^2\right)} ds\right) \\
&\leq D_\alpha(\pi_R^{T+K} \| \pi_U^K) \exp\left(\int_0^t -\frac{2\sigma^2}{\alpha\left((1 + \eta L)^2 C_{U,K} + 2s\sigma^2\right)} ds\right) \qquad (\text{Lemma A.2})
\end{aligned}$$

Computing the integral yields:

$$\begin{aligned}
\int_0^t -\frac{2\sigma^2}{\alpha\left((1 + \eta L)^2 C_{U,K} + 2s\sigma^2\right)} ds &= -\frac{1}{\alpha} \int_0^t \frac{2\sigma^2}{(1 + \eta L)^2 C_{U,K} + 2s\sigma^2} ds \\
&= -\frac{1}{\alpha}\left[\log\left((1 + \eta L)^2 C_{U,K} + 2t\sigma^2\right) - \log\left((1 + \eta L)^2 C_{U,K}\right)\right] \\
&= -\frac{1}{\alpha}\left[\log\left(1 + \frac{2t\sigma^2}{(1 + \eta L)^2 C_{U,K}}\right)\right]
\end{aligned}$$

Thus, by setting $t = \eta$, we get:

$$D_\alpha(\pi_R^{T+K,1,\eta} \| \pi_U^{K,1,\eta}) \leq \left(1 + \frac{2\eta\sigma^2}{(1+\eta L)^2 C_{U,K}}\right)^{\frac{-1}{\alpha}} D_\alpha(\pi_R^{T+K} \| \pi_U^K)$$

Finally, using the data processing inequality for the projection of PNGD and iterating over the number of unlearning iterations, we get:

$$\begin{aligned}
D_\alpha(\pi_R^{T+K+1} \| \pi_U^{K+1}) &\leq D_\alpha(\pi_R^{T+K,1,\eta} \| \pi_U^{K,1,\eta}) \\
&\leq \left(1 + \frac{2\eta\sigma^2}{(1+\eta L)^2 C_{U,K}}\right)^{\frac{-1}{\alpha}} D_\alpha(\pi_R^{T+K} \| \pi_U^K) \\
&\leq D_\alpha(\pi_R^T \| \pi_L^T) \prod_{k=1}^K \left(1 + \frac{2\eta\sigma^2}{(1+\eta L)^2 C_{U,k}}\right)^{\frac{-1}{\alpha}}
\end{aligned}$$

$\square$

### A.2. Tracking the log-Sobolev constants

For a generic, $L$-smooth non-convex loss function $\mathcal{L}$, one can derive the following recurrence relation, $\forall k \geq 1$ upper bounding the log-Sobolev constants:

$$\begin{aligned}
C_1 &\leq (1+\eta L)^2 C_0 + 2\eta\sigma^2 & \text{(Lemma A.1)} \\
C_2 &\leq (1+\eta L)^4 C_0 + (1+\eta L)^2 2\eta\sigma^2 + (1+\eta L)^2 \\
&\cdots \\
C_K &\leq (1+\eta L)^{2K} C_0 + 2\eta\sigma^2 \sum_{k=0}^{K-1} (1+\eta L)^2 \\
&\leq (1+\eta L)^{2K} C_0 + 2\eta\sigma^2 \frac{(1+\eta L)^{2K} - 1}{(1+\eta L)^2 - 1} & (3)
\end{aligned}$$

If we add the assumption that the loss is convex, then the map $h(\theta) = \theta - \eta\nabla_\theta \mathcal{L}(\theta)$ is 1-Lipschitz for $\eta < \frac{2}{L}$ (Hardt et al., 2016) and we can reduce $(1+\eta L)$ to 1 in the aforementioned bounds:

$$C_K \leq C_0 + 2K\eta\sigma^2 \tag{4}$$

Finally, assuming $m$-strong convexity yields that the map $h(\theta)$ is $1 - \eta m$-Lipschitz, which allows for the following **contractive** recurrence on the log-Sobolev constants $\forall k \geq 1$ by setting $\eta < \frac{2}{m}(1 - \frac{\sigma^2}{mC_0})$ (Chien et al., 2024a):

$$\begin{aligned}
C_k &\leq (1-\eta m)^2 C_{k-1} + 2\eta\sigma^2 \leq C_{k-1} \\
C_k &\leq (1-\eta m)^{2K} C_0 + 2\eta\sigma^2 \frac{(1-\eta m)^{2K} - 1}{(1-\eta m)^2 - 1} \leq C_0
\end{aligned}$$

Thus, we have that $\forall t \in [0, \eta]$, $\pi_U^{K,1,t}$ satisfies a $C_0$ log-Sobolev inequality thus we have by Lemma A.4:

$$\frac{\partial D_\alpha(\pi_R^{T+K,1,t} \| \pi_U^{K,1,t})}{\partial t} \leq -\frac{2\sigma^2}{\alpha C} D_\alpha(\pi_R^{T+K,1,t} \| \pi_U^{K,1,t})$$

Thus, by Grönwall's inequality (Lemma A.5), we have $\forall t \in [0, \eta]$:

$$\begin{aligned}
\frac{\partial D_\alpha(\pi_R^{T+K,1,t} \| \pi_U^{K,1,t})}{\partial t} &\leq D_\alpha(h_\sharp \pi_R^{T+K} \| h_\sharp \pi_U^K) \exp\left(\int_0^t -\frac{2\sigma^2}{\alpha C} ds\right) \\
&\leq D_\alpha(h_\sharp \pi_R^{T+K} \| h_\sharp \pi_U^K) \exp\left(-\frac{2t\sigma^2}{\alpha C}\right)
\end{aligned}$$

Thus, by setting $t = \eta$ and using similar steps as the non convex proof above, we get the following result:

$$D_\alpha(\pi_R^{T+K} \| \pi_U^K) \leq D_\alpha(\pi_R^T \| \pi_L^T) \exp\left(-\frac{2K\eta\sigma^2}{\alpha C}\right)$$

The message conveyed by the strongly convex proof is that if we have a universal iteration independent upper bound on the log Sobolev constants at each timestep of the PNGD updates, then we could have a more meaningful upper bound on the Rényi divergence. The non convex Equation (3) and convex Equation (4) recurrence bounds are non contractive and iteration dependent, so they do not allow to establish a convergence rate for Theorem 3.2. This is where the projection step of PNGD comes in handy, as it allows to leverage the geometry of the set $\Theta$ to get a more informative bound:

**Lemma A.7** (Log Sobolev inequality on measures supported on a compact set (Chen et al., 2021), Corollary 1). *Let $\pi$ be a probability measure on $\mathbb{R}^d$ supported on a compact set $\Theta$ with radius $R \geq 0$. Then, for each $t \geq 0$, $\mu * \mathcal{N}(0, tI_d)$ satisfy a log sobolev inequality with constant $C$ controlled by:*

$$C \leq 6\left(4R^2 + t\right)\exp\left(\frac{4R^2}{t}\right)$$

**Proposition A.1** (Universal bound on the log Sobolev constants of distributions induced by PNGD updates (Chien et al., 2024a)). *Suppose that $\mathcal{L}$ is $M$ Lipschitz. Let $\theta_0 \sim \pi_0 \in \mathcal{P}(\Theta)$ where $\Theta$ is a compact set of radius $R$ and denote by $\pi_k$ the distribution $\theta_k$, the $k$-th iterate of PNGD (Equation (1)). Then, $\forall k \geq 0$, $\pi_k$ satisfies a log-Sobolev inequality with constant $C_k$ controlled by:*

$$C_k \leq 6\left(4(R + \eta M)^2 + 2\eta\sigma^2\right)\exp\left(\frac{4(R + \eta M)^2}{2\eta\sigma^2}\right)$$

We can thus derive a similar bound to the strongly convex setting, for the non convex/convex settings:

Using Proposition A.1, we have $\forall k \geq 0$ that $\pi_U^K$ satisfies a $\tilde{C} = 6\left(4(R + \eta M)^2 + 2\eta\sigma^2\right)\exp\left(\frac{4(R+\eta M)^2}{2\eta\sigma^2}\right)$ log Sobolev inequality. Thus, using Lemma A.4, we have:

$$\frac{\partial D_\alpha(\pi_R^{T+K,1,t} \| \pi_U^{K,1,t})}{\partial t} \leq -\frac{2\sigma^2}{\alpha\tilde{C}} D_\alpha(\pi_R^{T+K,1,t} \| \pi_U^{K,1,t})$$

Thus, by Grönwall's inequality (Lemma A.5), we have $\forall t \in [0, \eta]$:

$$\frac{\partial D_\alpha(\pi_R^{T+K,1,t} \| \pi_U^{K,1,t})}{\partial t} \leq D_\alpha(h_\sharp \pi_R^{T+K} \| h_\sharp \pi_U^K) \exp\left(\int_0^t -\frac{2\sigma^2}{\alpha\tilde{C}} ds\right)$$
$$\leq D_\alpha(h_\sharp \pi_R^{T+K} \| h_\sharp \pi_U^K) \exp\left(-\frac{2t\sigma^2}{\alpha\tilde{C}}\right)$$

Finally, similarly to the strongly convex proofs, we can deduce that:

$$D_\alpha(\pi_R^{T+K} \| \pi_U^K) \leq D_\alpha(\pi_R^T \| \pi_L^T) \exp\left(-\frac{2K\eta\sigma^2}{\alpha\tilde{C}}\right)$$

# B. On training with public data

## B.1. Proof of Theorem 3.1

**Theorem 3.1** (The role of public data in shrinking the learning / retraining mismatch.). *Suppose that the loss is $L$-smooth and $M$-Lipschitz, and that the initialization distribution satsifies a $C_0$-log Sobolev inequality. Moreover, suppose that the PNGD updates project onto a compact set $\Theta$ of radius $R$.*
*Then at learning iteration T, we have the following upper bound on the Rényi divergence between the retraining $\pi_R^T$ and learning $\pi_L^T$ distributions:*

$$\frac{D_\alpha(\pi_R^T \| \pi_L^T)}{\alpha} \leq \frac{2M^2\eta^2 n_{\text{forget}}^2}{(n_{\text{pub}} + n_{\text{priv}})^2\sigma^2} \sum_{t=1}^{T-1} \prod_{t'=t}^{T-1} h(t', \eta, \sigma),$$

*where* $h(t', \eta, \sigma) = \left(1 + \frac{\eta\sigma^2}{C_{t',1}}\right)^{-1}$, *and* $0 < C_{t',1} \le (1+\eta L)^{2K} C_0 + 2\eta\sigma^2 \frac{(1+\eta L)^{2K}-1}{(1+\eta L)^2-1}$ *are log Sobolev constants of the distributions of the intermediate PNGD updates. Using the support's radius allows to loosely upper bound those constants* (*Chien et al., 2024a*): $C_{t',1} \le 6e^{\frac{4\tau}{\eta\sigma^2}}(4\tau^2 + \eta\sigma^2)$ *with* $\tau = R + \eta M$.

*Proof.* The following proof is an adaptation of the proof of Theorem 3.2 in Chien et al. (2024a) to the asymmetric data setting.

Consider the following updates done during training. Recall that we are using full batch projected noisy gradient descent:

$$\theta_L^{t+1} = \Pi_\Theta \left[ \theta_L^t + \eta \nabla \mathcal{L}_{D_{\mathrm{pub}} \cup D_{\mathrm{priv}}}(\theta_L^t) + \sqrt{2\eta\sigma^2}W_t \right] \qquad (W_t \sim \mathcal{N}(0, I_d))$$

$$\theta_R^{t+1} = \Pi_\Theta \left[ \theta_R^t + \eta \nabla \mathcal{L}_{D_{\mathrm{retain}}}(\theta_R^t) + \sqrt{2\eta\sigma^2}W_t \right] \qquad (W_t \sim \mathcal{N}(0, I_d))$$

Let's divide each optimization step into the following:

$$\theta_L^{t,1} = \theta_L^t + \eta \nabla \mathcal{L}_{D_{\mathrm{pub}} \cup D_{\mathrm{priv}}}(\theta_L^t) + \sqrt{\eta\sigma^2}W_t$$

$$\theta_R^{t,1} = \theta_R^t + \eta \nabla \mathcal{L}_{D_{\mathrm{retain}}}(\theta_R^t) + \sqrt{\eta\sigma^2}W_t$$

Therefore, we can write

$$\theta_L^{t+1} = \Pi_\Theta \left[ \theta_L^{t,1} + \sqrt{\eta\sigma^2}W_t \right] \tag{5}$$

$$\theta_R^{t+1} = \Pi_\Theta \left[ \theta_R^{t,1} + \sqrt{\eta\sigma^2}W_t \right]. \tag{6}$$

Let $\pi_R^t, \pi_R^{t,1}, \pi_L^t, \pi_L^{t,1}$ be the distributions of respectively $\theta_R^t, \theta_R^{t,1}, \theta_L^t, \theta_L^{t,1}$

**The main question we try to tackle here is: what is $D_\alpha(\pi_\mathbf{R}^\mathbf{t} \| \pi_\mathbf{L}^\mathbf{t})$ ?**
We first compare the distributions $\pi_R^{t,1}$ and $\pi_L^{t,1}$. By composition theorem of the Gaussian mechanism for Rényi Differential privacy (Mironov, 2017), and equivalently for the Rényi divergence, we have:

$$\frac{D_\alpha(\pi_R^{t,1} \| \pi_L^{t,1})}{\alpha} \le \frac{D_\alpha(\pi_R^t \| \pi_L^t)}{\alpha} + \frac{\Delta_F^2}{2\sigma^2} \tag{7}$$

where $\Delta_F$ is the $l_2$ sensitivity of the gradient update. For the next computations, let $n_{\mathrm{pub}}$ denote the number of public points, $n_{\mathrm{forget}}$ denote the number of points to forget, and $n_{\mathrm{r-priv}}$ denote the number of *remaining* private points in the retain set. Computing the sensitivity in the asymmetric setting yields:

$$\Delta_F = \max_\theta \eta \|\nabla \mathcal{L}_{D_{\mathrm{retain}}}(\theta) - \nabla \mathcal{L}_{D_{\mathrm{pub}} \cup D_{\mathrm{priv}}}(\theta)\|$$

$$= \max_\theta \eta \| \frac{1}{n_{\mathrm{pub}} + n_{\mathrm{r-priv}}} \sum_{d_i \in \mathrm{I} \cup \mathrm{II}} \nabla \ell(\theta, d_i) - \frac{1}{n_{\mathrm{pub}} + n_{\mathrm{r-priv}} + n_{\mathrm{forget}}} \sum_{d_i \in \mathrm{I} \cup \mathrm{II} \cup \mathrm{III}} \nabla \|\ell(\theta, d_i)\| $$

$$\le \eta \left( \frac{1}{n_{\mathrm{pub}} + n_{\mathrm{r-priv}}} - \frac{1}{n_{\mathrm{pub}} + n_{\mathrm{r-priv}} + n_{\mathrm{forget}}} \right) \sum_{d_i \in \mathrm{I} \cup \mathrm{II}} \|\nabla \ell(\theta, d_i)\|$$

$$+ \frac{\eta}{n_{\mathrm{pub}} + n_{\mathrm{r-priv}} + n_{\mathrm{forget}}} \sum_{d_i \in \mathrm{I} \cup \mathrm{II} \cup \mathrm{III}} \|\nabla \ell(\theta, d_i)\|$$

$$\le M\eta(n_{\mathrm{pub}} + n_{\mathrm{r-priv}}) \left( \frac{1}{n_{\mathrm{pub}} + n_{\mathrm{r-priv}}} - \frac{1}{n_{\mathrm{pub}} + n_{\mathrm{r-priv}} + n_{\mathrm{forget}}} \right) + \frac{n_{\mathrm{forget}} M\eta}{n_{\mathrm{pub}} + n_{\mathrm{r-priv}} + n_{\mathrm{forget}}}$$

$$\le \underbrace{\frac{2M\eta n_{\mathrm{forget}}}{n_{\mathrm{pub}} + n_{\mathrm{r-priv}} + n_{\mathrm{forget}}}}_{\varepsilon}$$

**Lemma B.1.** *(Ye & Shokri, 2022) For any distributions $\xi_t, \xi_t'$ both satisfying $C_{t,1}$-LSI, we have:*

$$\frac{D_\alpha(\xi_t * \mathcal{N}(0, \eta\sigma^2 I), \xi_t' * \mathcal{N}(0, \eta\sigma^2 I))}{\alpha} \le \frac{D_{\alpha(t)}(\xi_t, \xi_t')}{\alpha(t)} \left(1 + \frac{\eta\sigma^2}{C_{t,1}}\right)^{-1}$$

*where* $\alpha(t) = \frac{\alpha - 1}{1 + \frac{\eta\sigma^2}{C_{t,1}}}$

By combining the data processing inequality (projection) and Lemma B.1, we get the following recurrence inequality:

$$
\begin{aligned}
\frac{D_\alpha(\pi_R^{T+1} \| \pi_L^{T+1})}{\alpha} &\leq \left( \frac{D_{\alpha(T)}(\pi_R^T \| \pi_L^T)}{\alpha(T)} + \frac{\varepsilon^2}{2\sigma^2} \right) \left( 1 + \frac{\eta\sigma^2}{C_{T,1}} \right)^{-1} \\
&= \frac{\varepsilon^2}{2\sigma^2} \left( 1 + \frac{\eta\sigma^2}{C_{T,1}} \right)^{-1} + \frac{D_{\alpha(T)}(\pi_R^T \| \pi_L^T)}{\alpha(T)} \left( 1 + \frac{\eta\sigma^2}{C_{T,1}} \right)^{-1} \\
&\leq \frac{\varepsilon^2}{2\sigma^2} \left( 1 + \frac{\eta\sigma^2}{C_{T,1}} \right)^{-1} + \left( \frac{D_{\alpha(T-1)}(\pi_R^T \| \pi_L^T)}{\alpha(T-1)} + \frac{\varepsilon^2}{2\sigma^2} \right) \left( 1 + \frac{\eta\sigma^2}{C_{T,1}} \right)^{-1} \left( 1 + \frac{\eta\sigma^2}{C_{T-1,1}} \right)^{-1} \\
&\leq \frac{\varepsilon^2}{2\sigma^2} [B(T) + B(T-1)] + B(T-2) \left( \frac{D_{\alpha(T-2)}(\pi_R^T \| \pi_L^T)}{\alpha(T-2)} + \frac{\varepsilon^2}{2\sigma^2} \right)
\end{aligned}
$$

$$
\left( \text{where } B(t) = \prod_{k=t}^{T} \left( 1 + \frac{\eta\sigma^2}{C_{k,1}} \right)^{-1} \right)
$$

$$
\begin{aligned}
&\leq \frac{\varepsilon^2}{2\sigma^2} \sum_{i=1}^{T} B(i) + B(0) \left( \frac{D_{\alpha(0)}(\pi_R^T \| \pi_L^T)}{\alpha(0)} + \frac{\varepsilon^2}{2\sigma^2} \right) \\
&\leq \frac{\varepsilon^2}{2\sigma^2} \sum_{i=0}^{T} B(i) && \text{(since } D_{\alpha(t)}(\pi_0 \| \pi_0) = 0) \\
&= \frac{\varepsilon^2}{2\sigma^2} \sum_{t=0}^{T} \prod_{t'=t}^{T} \left( 1 + \frac{\eta\sigma^2}{C_{t',1}} \right)^{-1}
\end{aligned}
$$

The upper bound on the log Sobolev constants can be tracked in a similar fashion as in Proposition A.1 because of the projection onto the compact set $\Theta$. $\qquad \square$

Corollary 3.1 follows immediately from the previous theorem:

**Corollary 3.1** (LU noise vs ALU noise). *Consider the setting where the forget set size constitutes a constant fraction of the private data (i.e., $n_{\text{forget}} = c n_{\text{priv}}$ for some $c \in (0,1]$). Let $\sigma^2_{\text{sym}}$ and $\sigma^2_{\text{asym}}$ be the noise variances required to bound the Rényi divergence between the learning and retraining distributions by a fixed $\varepsilon$ in the symmetric and asymmetric settings, respectively. These variances satisfy the following lower bounds:*

$$
\sigma^2_{\text{sym}} \geq \frac{2M^2 \eta^2 c^2}{\varepsilon} \sum_{t=1}^{T-1} \prod_{t'=t}^{T-1} h(t', \eta, \sigma),
$$

$$
\sigma^2_{\text{asym}} \geq \frac{2M^2 \eta^2 c^2}{\varepsilon} \left( \sum_{t=1}^{T-1} \prod_{t'=t}^{T-1} h(t', \eta, \sigma) \right) \left( \frac{n_{\text{priv}}}{n_{\text{priv}} + n_{\text{pub}}} \right)^2.
$$

Starting with the bound of Theorem 3.1, we have:

$$
D_\alpha(\pi_R^T \| \pi_L^T) \leq \frac{2\alpha M^2 \eta^2 n_{\text{forget}}^2}{(n_{\text{pub}} + n_{\text{priv}})^2 \sigma^2} \sum_{t=1}^{T-1} \prod_{t'=t}^{T-1} h(t', \eta, \sigma).
$$

## B.2. Proof of Corollary 3.1

One sufficient condition to satisfy an upper bound $\varepsilon$ on the Rényi divergence is:

$$
\frac{2\alpha M^2 \eta^2 n_{\text{forget}}^2}{(n_{\text{pub}} + n_{\text{priv}})^2 \sigma^2} \sum_{t=1}^{T-1} \prod_{t'=t}^{T-1} h(t', \eta, \sigma) \leq \varepsilon
$$

$$
\iff \frac{2\alpha M^2 \eta^2 n_{\text{forget}}^2}{(n_{\text{pub}} + n_{\text{priv}})^2 \varepsilon} \sum_{t=1}^{T-1} \prod_{t'=t}^{T-1} h(t', \eta, \sigma) \leq \sigma^2.
$$

Since $\frac{n_{\text{forget}}}{n_{\text{public}}+n_{\text{priv}}} = \frac{n_{\text{forget}}}{n_{\text{priv}}}\frac{n_{\text{priv}}}{n_{\text{public}}+n_{\text{priv}}}$, we set $c = \frac{n_{\text{forget}}}{n_{\text{priv}}}$ and can thus write:

$$\sigma^2 \geq \frac{2\alpha M^2 \eta^2 c^2}{\varepsilon}(\sum_{t=1}^{T-1}\prod_{t'=t}^{T-1} h(t',\eta,\sigma))\left(\frac{n_{\text{priv}}}{n_{\text{public}}+n_{\text{priv}}}\right)^2 \tag{8}$$

When the loss is $m$-strongly convex, we can also derive a right hand side that is completely independent of $\sigma$:

$$\sigma^2 \geq \frac{4\alpha c^2 M^2(1-\exp(-m\eta T))}{\varepsilon m}\left(\frac{n_{\text{priv}}}{n_{\text{public}}+n_{\text{priv}}}\right)^2 \tag{9}$$

Compressed Version

### B.3. Numerical Setup for Strongly Convex Losses

Following Chien et al. (2024a), we evaluate our bounds using binary logistic regression on $D = \{(x_i, y_i)\}_{i=1}^n$:

$$\mathcal{L}(\theta, D) = -\frac{1}{n}\sum_{i=1}^n \log\left(s(y_i\theta^T x_i)\right) + \frac{\lambda}{2}|\theta|_2^2$$

where $s(\cdot)$ is the sigmoid function. To ensure $M$-Lipschitzness, gradients are clipped during training, which preserves the $\lambda$-strong convexity and $(\frac{1}{4}+\lambda)$-smoothness of the objective (Ye & Shokri, 2022).For the results in Figure 2, we adopt the MNIST configuration from Chien et al. (2024a): $\lambda = 0.0119$, $T = 10^4$, $n_{\text{priv}} = 3000$, $M = 1$, $\alpha = 2$, and $\eta = 1/L$. We set the target privacy budget to $\varepsilon = 1$ and vary the forget set fraction to compute Equation (9).

## C. Proof of Theorem 4.1

**Proposition.** *Assuming the data generating distributions share the same support, that the weight space $\Theta$ is compact and that the loss is $M$-Lipschitz wrt $\theta$, we have the following upper bound on the generalization error on the private data after performing $K$ iterations of unlearning, and initializing a weight $\theta_0$ from $\pi_L^T$:*

$$\mathbb{E}_{\theta\sim\pi_U}\left[\mathbb{E}_{x\sim P_{\text{priv}}}[\mathcal{L}(\theta,x)]\right] \leq \underbrace{\exp\left(\frac{n_{\text{pub}}}{n_{\text{pub}}+n_{\text{retain}}}D_\infty(P_{\text{priv}}\|P_{\text{pub}})\right)}_{\text{distribution mismatch penalty}}\mathbb{E}_{\theta\sim\pi_R}\left[\mathbb{E}_{d\sim P_{\text{train}}}[\mathcal{L}(\theta,d)]\right] +$$

$$M \times diam(\Theta) \times \underbrace{\sqrt{\frac{1}{2}D_\alpha(\pi_R\|\pi_U)}}_{\text{unlearning approximation error}}$$

*where $D_\infty(P\|Q) = \log\left(\text{ess sup}_{x\sim Q}\frac{p(x)}{q(x)}\right)$ is the infinite Rényi divergence (worst case regret (Erven & Harremoës, 2014)) and $p_{\text{train}}$ denotes the mixture of distributions $D_{\text{pub}}$ and $D_{\text{priv}}$ used for training the model.*

In order to prove Theorem 4.1, we will use the following quantities to define a set of preliminary lemmas.

C.0.1. PERFORMANCE ON THE TRAINING DISTRIBUTION MIXTURE

**Definition C.1** (Wasserstein distance). *The Wasserstein-1 distance is defined as*

$$W_1(\mu,\nu) = \inf_{\gamma\in\Pi(\mu,\nu)}\int_{\mathcal{X}\times\mathcal{X}}d(x,y)\,d\gamma(x,y),$$

*where:*

- *$\mu$ and $\nu$ are probability measures on a metric space $(\mathcal{X}, d)$,*

- *$d(x,y)$ is the distance between points $x, y \in \mathcal{X}$,*

- $\Pi(\mu, \nu)$ is the set of all couplings of $\mu$ and $\nu$, i.e., the set of joint distributions $\gamma$ on $\mathcal{X} \times \mathcal{X}$ such that the marginals of $\gamma$ are $\mu$ and $\nu$:

$$\int_{\mathcal{X}} \gamma(x, y)\, dy = \mu(x), \quad \int_{\mathcal{X}} \gamma(x, y)\, dx = \nu(y).$$

**Definition C.2** (Total Variation Distance). *Let $P$ and $Q$ be two probability measures on a measurable space $(\Omega, \mathcal{F})$. The total variation distance between $P$ and $Q$ is defined as*

$$TV(P, Q) = \sup_{A \in \mathcal{F}} |P(A) - Q(A)| \tag{10}$$

$$= \frac{1}{2} \int_{\Omega} |dP - dQ| \tag{11}$$

$$= \frac{1}{2} \|P - Q\|_{TV}. \tag{12}$$

**Theorem C.1.** *(Kantorovich Rubinstein's duality, (Villani et al., 2009), Theorem 5.10) If $\mu, \nu$ have a bounded support $\Omega$, then*

$$W_1(\mu, \nu) = \sup_{\|h\|_L \leq 1} \mathbb{E}_{x \sim \mu}[h(x)] - \mathbb{E}_{y \sim \nu}[h(y)], \tag{13}$$

*where $\|h\|_L \leq 1$ denotes the set of 1-Lipschitz functions on $\Omega$*

Let $f : \Theta \to \mathbb{R}$ such that $f(\theta) = \mathbb{E}_{D \sim P_{\text{train}}}[\mathcal{L}_D(\theta)]$, where $P_{\text{train}}$ denotes the training data distribution (a mixture of $P_{\text{priv}}$ and $P_{\text{pub}}$. Since $\mathcal{L}(., D)$ is $M-$Lipschitz, so is $f$. Then, we have that:

$$\mathbb{E}_{\substack{\theta \sim \pi_U \\ D \sim P_{\text{train}}}} [\mathcal{L}(\theta, D)] - \mathbb{E}_{\substack{\theta \sim \pi_R \\ D \sim P_{\text{train}}}} [\mathcal{L}(\theta, D)] = \mathbb{E}_{\theta \sim \pi_U}[f(\theta)] - \mathbb{E}_{\theta \sim \pi_R}[f(\theta)] \qquad \text{(Fubini's theorem)}$$

$$\leq M \times W_1(\pi_U, \pi_R) \qquad \text{(By Theorem C.1)}$$

Now, we need to find an upper bound on the 1-Wasserstein distance in terms of the Rényi divergence between $\pi_R$ and $\pi_U$. The following results will be useful in deriving it:

**Proposition C.1.** *(Pinsker's inequality) For two probability distributions $P, Q$, we have*

$$2TV(P, Q)^2 \leq KL(P\|Q). \tag{14}$$

**Proposition C.2.** *(Monotonicity of Rényi divergence, (Erven & Harremoës, 2014)) For $1 \leq \alpha_1 \leq \alpha_2$ and probability measures $P, Q$,*

$$KL(P\|Q) \leq D_{\alpha_1}(P\|Q) \leq D_{\alpha_2}(P\|Q).$$

*The KL lower bounds any Rényi divergence since it is obtained by the limit $\alpha \to 1$.*

**Proposition C.3.** *(Upper bounding $W_1$ with $TV$ (Gibbs & Su, 2002)) If the distributions $P, Q$ share a support $\Omega$ and $diam(\Omega) = \sup_{(x,y) \in \Omega \times \Omega} d(x, y)$ is finite, then we have*

$$W_1(P, Q) \leq diam(\Omega) TV(P, Q). \tag{15}$$

Using the results above, we have

$$\mathbb{E}_{\theta \sim \pi_U}[f(\theta)] - \mathbb{E}_{\theta \sim \pi_R}[f(\theta)] \leq M W_1(\pi_U, \pi_R)$$

$$\leq M \times diam(\Theta) \times TV(\pi_U, \pi_R) \qquad \text{(By Proposition C.3 and compactness of } \Theta)$$

$$\leq M \times diam(\Theta) \times \sqrt{\frac{1}{2} KL(\pi_U, \pi_R)} \qquad \text{(By Proposition C.2)}$$

$$\leq M \times diam(\Theta) \times \sqrt{\frac{1}{2} D_\alpha(\pi_U, \pi_R)} \qquad \text{(By Proposition C.2)}$$

Thus, we obtain that the generalization error of learning + unlearning is upper bounded by:

**Proposition C.4.** *Assuming that $\mathcal{L}$ is $M$-Lipschitz, we have*

$$\mathbb{E}_{\theta \sim \pi_U} [\mathbb{E}_{D \sim P_{\text{train}}}[L(\theta, D)]] \leq \mathbb{E}_{\theta \sim \pi_R} [\mathbb{E}_{D \sim P_{\text{train}}}[L(\theta, D)]] + M \times diam(\Theta) \times \sqrt{\frac{1}{2} D_\alpha(\pi_U \| \pi_R)} \tag{16}$$

C.0.2. ADAPTING THE BOUND TO THEOREM 4.1

We would like to evaluate the performance of the model obtained after unlearning. Proposition C.4 provides a generalization bound on a mixture of distributions, namely on public data + private data. In most practical scenarios, one would want to quantify the "lost" performance on private data after forgetting one of its subsets. Thus, we would like to upper bound the quantity $\mathbb{E}_{\pi_U}\left[\mathbb{E}_{D \sim P_{\text{priv}}}[\mathcal{L}_D(\theta)]\right]$. The training data distribution used for either retraining or unlearning can be considered as generated from a mixture of the distributions $I$ and $II$. Assuming the sampling proportions for training are consistent, one can write that the data distribution used in retraining is

$$P_{\text{train}} = \frac{n_{\text{pub}}}{n_{\text{pub}} + n_{\text{r-priv}}} P_{\text{pub}} + \frac{n_{\text{r-priv}}}{n_{\text{pub}} + n_{\text{r-priv}}} P_{\text{priv}}.$$

Fix any $\theta \in \Theta$. We have that

$$\mathbb{E}_{\mathcal{D} \sim P_{\text{train}}}[\mathcal{L}(\theta, \mathcal{D})] = \frac{n_{\text{pub}}}{n_{\text{pub}} + n_{\text{r-priv}}} \mathbb{E}_{\mathcal{D} \sim P_{\text{pub}}}[\mathcal{L}(\theta, \mathcal{D})] + \frac{n_{\text{r-priv}}}{n_{\text{pub}} + n_{\text{r-priv}}} \mathbb{E}_{\mathcal{D} \sim P_{\text{priv}}}[\mathcal{L}(\theta, \mathcal{D})]$$

$$
\begin{aligned}
\mathbb{E}_{\mathcal{D} \sim P_{\text{priv}}}[\mathcal{L}(\theta, \mathcal{D})] &= \int p_{\text{priv}}(x) \mathcal{L}(\theta, x) dx \\
&= \int p_{\text{train}}(x) \frac{p_{\text{priv}}(x)}{p_{\text{train}}(x)} \mathcal{L}(\theta, x) dx \\
&= \mathbb{E}_{x \sim P_{\text{train}}}\left[\frac{p_{\text{priv}}(x)}{p_{\text{train}}(x)} \mathcal{L}(\theta, x)\right] \\
&\leq \mathbb{E}_{d \sim P_{\text{train}}}\left[\operatorname{ess\,sup}_{x \in Supp(P_{\text{pub}}) \cup Supp(P_{\text{priv}})} \frac{p_{p_{\text{priv}}}(x)}{p_{\text{train}}(x)} \mathcal{L}(\theta, d)\right] \\
&\leq \operatorname{ess\,sup}_{x \in Supp(P_{\text{pub}}) \cup Supp(P_{\text{priv}})} \frac{p_{\text{priv}}(x)}{p_{\text{train}}(x)} \mathbb{E}_{d \sim P_{\text{train}}}[\mathcal{L}(\theta, d)] \\
&\leq \exp(D_\infty(P_{\text{priv}}, P_{\text{train}})) \mathbb{E}_{d \sim P_{\text{train}}}[\mathcal{L}(\theta, d)].
\end{aligned}
$$

Moreover, we have by convexity of the Rényi divergence (Erven & Harremoës, 2014) in its second argument that

$$D_\infty(P_{\text{priv}} \| P_{\text{train}}) \leq \frac{n_{\text{pub}}}{n_{\text{pub}} + n_{\text{r-priv}}}(P_{\text{priv}} \| P_{\text{pub}}).$$

Thus we also have

$$\mathbb{E}_{d \sim P_{\text{priv}}}[\mathcal{L}(\theta, d)] \leq \exp\left(\frac{n_{\text{pub}}}{n_{\text{pub}} + n_{\text{r-priv}}} D_\infty(P_{\text{priv}} \| P_{\text{pub}})\right) \mathbb{E}_{d \sim P_{\text{train}}}[\mathcal{L}(\theta, d)]. \tag{17}$$

Thus, we can adapt proposition C.4 to evaluate the risk *only* on private data. Note that so far, the only assumption made on the difference between the data generating distributions I and II is that they share the same support. The following bound might be refined with additional assumptions, such as covariate shift or conditional shift.

We can thus take the expectation of $\theta$ with respect to $\pi_U$ to get

$$\mathbb{E}_{\theta \sim \pi_U}\left[\mathbb{E}_{d \sim P_{\text{priv}}}[\mathcal{L}(\theta, d)]\right] \leq \exp\left(\frac{n_{\text{pub}}}{n_{\text{pub}} + n_{\text{r-priv}}} D_\infty(P_{\text{priv}} \| P_{\text{pub}})\right) \mathbb{E}_{\theta \sim \pi_U}\left[\mathbb{E}_{d \sim P_{\text{train}}}[\mathcal{L}(\theta, d)]\right],$$

and using proposition C.4 to upper bound $\mathbb{E}_{\theta \sim \pi_U}\left[\mathbb{E}_{d \sim P_{\text{train}}}[\mathcal{L}(\theta, d)]\right]$, we prove proposition 4.1:

$$\mathbb{E}_{\theta \sim \pi_U}\left[\mathbb{E}_{x \sim P_{\text{priv}}}[\mathcal{L}(\theta, x)]\right] \leq \exp\left(\frac{n_{\text{pub}}}{n_{\text{pub}} + n_{\text{retain}}} D_\infty(P_{\text{priv}} \| P_{\text{pub}})\right) \mathbb{E}_{\theta \sim \pi_R}\left[\mathbb{E}_{d \sim P_{\text{train}}}[\mathcal{L}(\theta, d)]\right] +$$

$$M \times diam(\Theta) \times \sqrt{\frac{1}{2} D_\alpha(\pi_R \| \pi_U)}.$$

### C.1. Retraining performance bound on the training error

($\mathbb{E}_{\theta \sim \pi_{\mathbf{R}}^{\mathbf{T}}}[\mathbb{E}_{\mathbf{x} \sim \mathbf{p}_{\text{train}}}[\mathcal{L}(\theta, \mathbf{x})]]$): The upper bound could be further improved to include the *optimal* distribution, i.e by linking $\mathbb{E}_{\theta \sim \pi_{\mathbf{R}}^{\mathbf{T}}}[\mathbb{E}_{\mathbf{x} \sim \mathbf{p}_{\text{train}}}[\mathcal{L}(\theta, \mathbf{x})]]$ to $\arg\min_{\pi \in \mathcal{P}(\mathbb{R}^d)} \mathbb{E}_{\theta \sim \pi}[\mathbb{E}_{\mathbf{x} \sim \mathbf{p}_{\text{train}}}[\mathcal{L}(\theta, \mathbf{x})]]$. However, standard generalization bounds for Langevin dynamics (Raginsky et al., 2017; Xu et al., 2018) do not directly apply to our setting due to the projection operator $\Pi_\Theta$ in the PNGD updates. These classical results focus on unconstrained non-convex optimization, whereas our bounded domain introduces additional complexity. The most relevant analysis we are aware of is Lamperski (2020), who study generalization properties of projected Stochastic Gradient Langevin Dynamics, though their work considers the infinite-data regime.

## D. When to unlearn, when to retrain ?

In the following section, we will assume that the loss function is $m$-strongly convex to simplify the computations. Non-convex analysis follows the same flavor, but is hardly interpretable due to the presence of the log-Sobolev constants. As a reminder of ALU, we remind the reader that (1) one learns the whole dataset through $T$ iterations of PNGD, (2) unlearns by fine-tuning on the retain set using the same PNGD updates. As opposed to (Chien et al., 2024a;b) that defines the retraining cost as the number of PNGD steps required to get $\varepsilon$-close to the stationary distribution of retraining from scratch, **we define the cost of retraining as $T$, i.e the same number of steps used to train the model prior to unlearning**. This is a more realistic setting in practice as practitioners may choose the number of training steps to be an arbitrary number. The cost of unlearning is defined as the minimum amount of unlearning iterations, $K$, so that $D_\alpha(\pi_R^{T+K} \| \pi_U^K) \leq \varepsilon$, for a given $\varepsilon > 0$.

The following result gives a sufficient condition on when ALU is more advantageous than retraining, under the assumption that the loss is $m$-strongly convex. Note that a similar but hard-to-interpret result could be derived for non-convex losses, due to the presence of the log-Sobolev constants.

**Proposition D.1** (Sufficient condition guaranteeing that ALU is more efficient than retraining). *Assume that the conditions of Theorem 3.2 are satisfied, and that the loss function is $m$-strongly convex. Then, a sufficient condition guaranteeing that unlearning is more advantageous than retraining is*

$$\frac{C\alpha}{2\sigma^2\eta} \log\left( \frac{4\alpha M^2 n_{\text{forget}}^2}{m\varepsilon\sigma^2(n_{\text{public}} + n_{\text{priv}})^2} \right) < T - \log\left(1 - \exp(-m\eta T)\right).$$

*Proof.* Using the results from Theorems 3.1 and 3.2, with the assumption that the loss is $m$-strongly convex, we have that:

$$D_\alpha(\pi_R^{T+K} \| \pi_U^K) \leq D_\alpha(\pi_R^T \| \pi_L^T) \exp\left( -\frac{2K\sigma^2\eta}{C\alpha} \right).$$

By setting the right hand side to be less than $\varepsilon$, we obtain a sufficient condition on $K$:

$$D_\alpha(\pi_R^T \| \pi_L^T) \exp\left( -\frac{2K\sigma^2\eta}{C\alpha} \right) \leq \varepsilon$$

$$\Rightarrow \frac{C\alpha \log\left( \frac{D_\alpha(\pi_R^T \| \pi_L^T)}{\varepsilon} \right)}{2\sigma^2\eta} \leq K.$$

Therefore, a sufficient condition for unlearning to be computationally more efficient than retraining is:

$$\frac{C\alpha \log\left( \frac{4\alpha M^2 n_{\text{forget}}^2}{\varepsilon m\sigma^2(n_{\text{public}} + n_{\text{priv}})^2} \right)}{2\sigma^2\eta} \leq T - \log(1 - \exp(-m\eta T)). \quad \text{(By Theorem 3.1 and strong convexity of the loss)}$$

$\square$

This bound contains many dependencies between the control knobs that practitioners have in hand thanks to ALU. We ask the following questions:

**Small number of learning iterations:**    To analyze the computational advantage in this regime, we examine the sufficient condition when $T$ is small. Using the first-order Taylor approximation $1 - \exp(-m\eta T) \approx m\eta T$ and noting that for small $T$, the term $\exp(-\frac{2\sigma^2 \eta T}{C\alpha}) \approx 1$, the inequality yields:

$$\log\left(\frac{4\alpha M^2 n_{\text{forget}}^2}{\varepsilon m\sigma^2(n_{\text{public}} + n_{\text{priv}})^2}\right) \leq \frac{2\sigma^2 \eta}{C\alpha}(T - \log(m\eta T))$$

$$\Rightarrow \left(\frac{n_{\text{priv}}}{n_{\text{public}} + n_{\text{priv}}} \times \frac{n_{\text{forget}}}{n_{\text{priv}}}\right)^2 \leq \frac{\varepsilon m\sigma^2}{4\alpha M^2}\left[\underbrace{\exp\left(\frac{2\sigma^2 \eta T}{C\alpha}\right)}_{\approx 1}\exp\left(-\frac{2\sigma^2 \eta \log(m\eta T)}{C\alpha}\right)\right]$$

$$\Rightarrow \left(\frac{n_{\text{priv}}}{n_{\text{public}} + n_{\text{priv}}} \times \frac{n_{\text{forget}}}{n_{\text{priv}}}\right)^2 \leq \frac{\varepsilon m\sigma^2}{4\alpha M^2}(m\eta T)^{\frac{-2\sigma^2 \eta}{C\alpha}}$$

Letting $c = \frac{n_{\text{forget}}}{n_{\text{priv}}}$ denote the fraction of private data that needs to be forgotten, we have:

$$\left(\frac{n_{\text{priv}}}{n_{\text{public}} + n_{\text{priv}}}\right)^2 \leq \frac{\varepsilon m\sigma^2(m\eta T)^{\frac{-2\sigma^2 \eta}{C\alpha}}}{4\alpha M^2 c^2}$$

$$\Rightarrow (\frac{n_{\text{public}}}{n_{\text{priv}}} + 1)^2 \geq \frac{4\alpha M^2 c^2}{\varepsilon m\sigma^2}(m\eta T)^{\frac{2\sigma^2 \eta}{C\alpha}}$$

$$\Rightarrow \frac{n_{\text{public}}}{n_{\text{priv}}} \geq \underbrace{\frac{2\sqrt{\alpha}M(m\eta)^{\frac{\sigma^2 \eta}{C\alpha}}}{m}}_{\text{Const.}} \times T^{\frac{\sigma^2 \eta}{C\alpha}} \times \frac{1}{\sqrt{\varepsilon}} \times \left(\frac{n_{\text{forget}}}{n_{\text{priv}}}\right) - 1$$

In the regime where $T$ is small, the feasibility of unlearning is subject to a trade-off between the forget ratio, $\varepsilon$, and the available public data. This relationship is formally captured by the inequality $\frac{n_{\text{public}}}{n_{\text{priv}}} + 1 \geq \mathcal{C} \cdot T^\beta \cdot \varepsilon^{-1/2} \cdot \frac{n_{\text{forget}}}{n_{\text{priv}}}$, where $\beta = \sigma^2 \eta / C\alpha$. In the private-only scenario where $n_{\text{public}} = 0$, the efficiency of unlearning is limited; the algorithm only remains faster than retraining if the fraction of data to be forgotten is sufficiently small to satisfy $\frac{n_{\text{forget}}}{n_{\text{priv}}} \leq \frac{\sqrt{\varepsilon}}{\mathcal{C}T^\beta}$. Under these conditions, attempting to unlearn a larger portion of the dataset or imposing a stricter privacy guarantee (smaller $\varepsilon$) forces the unlearning iterations $K$ to surpass the original training budget $T$, rendering retraining a more viable option. However, the introduction of public data ($n_{\text{public}} > 0$) provides a computational buffer that fundamentally alters this dynamic. By increasing the ratio $n_{\text{public}}/n_{\text{priv}}$, practitioners can compensate for high forget ratios or stringent privacy requirements, ensuring that the gradient updates remain stable and non-private enough to allow $K < T$.

**Large number of training iterations (training to convergence)**    In this case, (Chien et al., 2024a) proved that the computational benefit of unlearning against retraining has a non-negligible benefit, that scales as $\mathcal{O}(\log(n_{\text{total}}))$, with probability $1 - \frac{1}{R^d}$.

## E. Langevin Unlearning pseudo-code

---

**Algorithm 1** Training with Projected Noisy Gradient Descent (PNGD)

---

1: $\theta_0 \sim \pi_0$ {Sample from initialization distribution}
2: **for** $t = 0$ to $T - 1$ **do**
3:     $g_t \leftarrow \nabla_\theta L_D(\theta_t)$ {Compute gradient on full dataset}
4:     $\xi_t \sim \mathcal{N}(0, 2\eta\sigma^2 I_d)$ {Sample Gaussian noise}
5:     $\theta_{t+1} \leftarrow \Pi_\Theta[\theta_t - \eta g_t + \xi_t]$ {Update and project}
6: **end for**
7: **return** $\theta_T$

---

**Algorithm 2** Langevin Unlearning

1: $\theta_0^U \leftarrow \theta_T$ {Initialize from trained model}
2: **for** $k = 0$ to $K - 1$ **do**
3: $\quad g_k \leftarrow \nabla_\theta L_{D_{\text{retain}}}(\theta_k^U)$ {Compute gradient on retain set only}
4: $\quad \xi_k \sim \mathcal{N}(0, 2\eta\sigma^2 I_d)$ {Sample Gaussian noise}
5: $\quad \theta_{k+1}^U \leftarrow \Pi_\Theta[\theta_k^U - \eta g_k + \xi_k]$ {Update and project}
6: **end for**
7: **return** $\theta_K^U$

# F. DomainNet data

The following is a snippet of samples from the DomainNet dataset, where we extracted two domains, Clipart and Quickdraw. The classes are aggregated into 24 meta-classes Table 3, following (Peng et al., 2019).

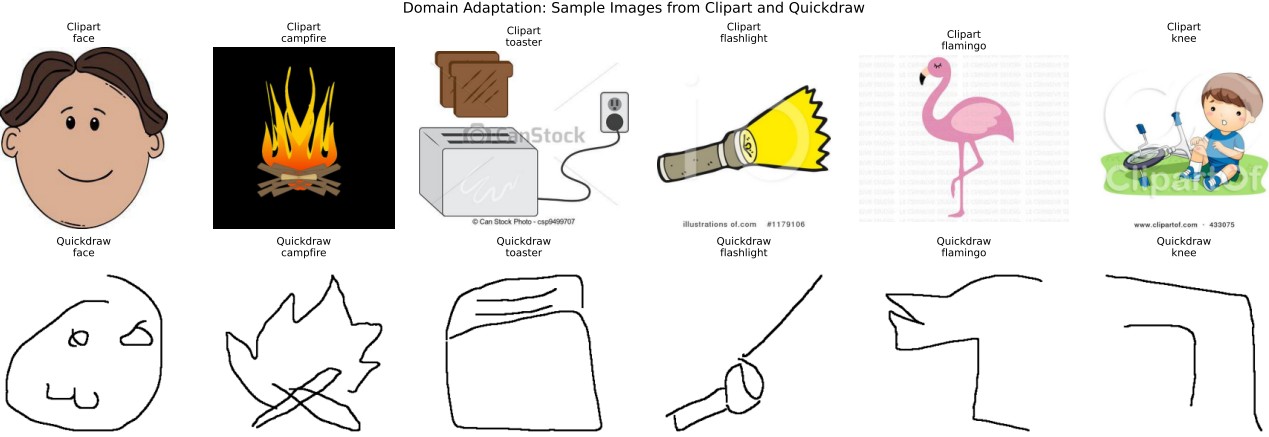

*Figure 5.* The two domains of public and private data used for Sections 5.1 and 5.2 (Peng et al., 2019). Both datasets share the same number of classes, with Clipart being a collection of stylized images representing the private data, and Quickdraw representing a collection of hand-draw sketches.

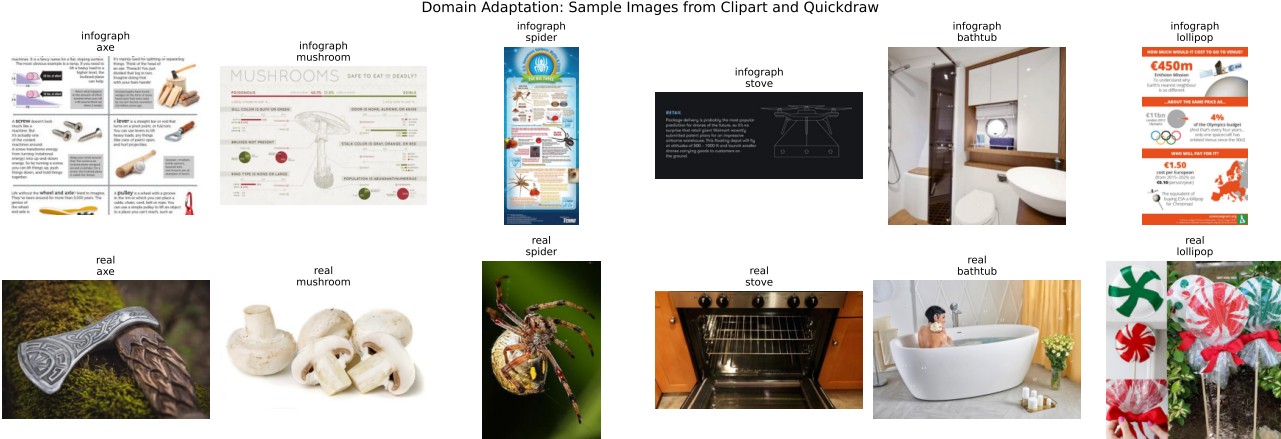

*Figure 6.* The two domains of public and private data used for Section 5.2 (Peng et al., 2019). Both datasets share the same number of classes, with Infograph being a collection of stylized images representing the public data, and Real representing a collection of real-life images.

*Table 3.* Class aggregation for experimental dataset. Individual classes are grouped into 24 superclasses.

| Superclass | Individual Classes |
| --- | --- |
| Furniture | bathtub, bed, bench, ceiling fan, chair, chandelier, couch, door, dresser, fence, fireplace, floor lamp, hot tub, ladder, lantern, mailbox, picture frame, pillow, postcard, see saw, sink, sleeping bag, stairs, stove, streetlight, suitcase, swing set, table, teapot, toilet, toothbrush, toothpaste, umbrella, vase, wine glass |
| Mammal | bat, bear, camel, cat, cow, dog, dolphin, elephant, giraffe, hedgehog, horse, kangaroo, lion, monkey, mouse, panda, pig, rabbit, raccoon, rhinoceros, sheep, squirrel, tiger, whale, zebra |
| Tool | anvil, axe, bandage, basket, boomerang, bottlecap, broom, bucket, compass, drill, dumbbell, hammer, key, nail, paint can, passport, pliers, rake, rifle, saw, screwdriver, shovel, skateboard, stethoscope, stitches, sword, syringe, wheel |
| Cloth | belt, bowtie, bracelet, camouflage, crown, diamond, eyeglasses, flip flops, hat, helmet, jacket, lipstick, necklace, pants, purse, rollerskates, shoe, shorts, sock, sweater, t-shirt, underwear, wristwatch |
| Electricity | calculator, camera, cell phone, computer, cooler, dishwasher, fan, flashlight, headphones, keyboard, laptop, light bulb, megaphone, microphone, microwave, oven, power outlet, radio, remote control, spreadsheet, stereo, telephone, television, toaster, washing machine |
| Building | The Eiffel Tower, The Great Wall, barn, bridge, castle, church, diving board, garden, garden hose, golf club, hospital, house, jail, lighthouse, pond, pool, skyscraper, square, tent, waterslide, windmill |
| Office | alarm clock, backpack, binoculars, book, calendar, candle, clock, coffee cup, crayon, cup, envelope, eraser, map, marker, mug, paintbrush, paper clip, pencil, scissors |
| Human Body | arm, beard, brain, ear, elbow, eye, face, finger, foot, goatee, hand, knee, leg, moustache, mouth, nose, skull, smiley face, toe, tooth |
| Road Transportation | ambulance, bicycle, bulldozer, bus, car, firetruck, motorbike, pickup truck, police car, roller coaster, school bus, tractor, train, truck, van |
| Food | birthday cake, bread, cake, cookie, donut, hamburger, hot dog, ice cream, lollipop, peanut, pizza, popsicle, sandwich, steak |
| Nature | beach, cloud, hurricane, lightning, moon, mountain, ocean, rain, rainbow, river, snowflake, star, sun, tornado |
| Cold Blooded | crab, crocodile, fish, frog, lobster, octopus, scorpion, sea turtle, shark, snail, snake, spider |
| Music | cello, clarinet, drums, guitar, harp, piano, saxophone, trombone, trumpet, violin |
| Fruit | apple, banana, blackberry, blueberry, grapes, pear, pineapple, strawberry, watermelon |
| Sport | baseball, baseball bat, basketball, flying saucer, hockey puck, hockey stick, snorkel, soccer ball, tennis racquet, yoga |
| Tree | bush, cactus, flower, grass, house plant, leaf, palm tree, tree |
| Bird | bird, duck, flamingo, owl, parrot, penguin, swan |
| Vegetable | asparagus, broccoli, carrot, mushroom, onion, peas, potato, string bean |
| Shape | circle, hexagon, line, octagon, squiggle, triangle, zigzag |
| Kitchen | fork, frying pan, hourglass, knife, lighter, matches, spoon, wine bottle |
| Water Transportation | aircraft carrier, canoe, cruise ship, sailboat, speedboat, submarine |
| Sky Transportation | airplane, helicopter, hot air balloon, parachute |
| Insect | ant, bee, butterfly, mosquito |
| Others | The Mona Lisa, angel, animal migration, campfire, cannon, dragon, feather, fire hydrant, mermaid, snowman, stop sign, teddy-bear, traffic light |

# G. Details about the Rényi estimation

### G.0.1. NEURAL RÉNYI ESTIMATION

Following the works of Birrell et al. (2021; 2023), two variational representations of the Rényi divergence between two distributions $P, Q$ have been proposed. The first draws inspiration from the Donsker–Varadhan dual representation (Donsker & Varadhan, 1975) of the KL divergence:

**Theorem G.1** (Donsker–Varadhan Rényi divergence (Birrell et al., 2021)). *Let $P, Q$ be two distributions on $(\Omega, \mathcal{M})$ and $\alpha \in \mathbb{R}$, $\alpha \neq 0, 1$. Then, for any set of functions $\Phi$ with $\mathcal{M}_b(\Omega) \subset \Phi \subset \mathcal{M}(\Omega)$,*

$$\frac{D_\alpha(P\|Q)}{\alpha} = \sup_{\phi \in \Phi} \left\{ \frac{1}{\alpha - 1} \log \int e^{(\alpha-1)\phi}\, dP - \frac{1}{\alpha} \log \int e^{\alpha\phi}\, dQ \right\}. \tag{18}$$

*If in addition $(\Omega, \mathcal{M})$ is a metric space with the Borel $\sigma$-algebra, then Equation (18) holds for all $\Phi$ satisfying $\mathrm{Lip}_b \subset \Phi \subset \mathcal{M}(\Omega)$, where $\mathrm{Lip}_b$ denotes the set of bounded Lipschitz functions.*

Here, $\mathcal{M}(\Omega)$ denotes the space of measurable real-valued functions on $\Omega$, and $\mathcal{M}_b(\Omega)$ the subspace of bounded functions.

While this representation allows sample-based estimation, it involves exponential terms that yield high-variance estimates in practice. To mitigate this issue, Birrell et al. (2023) proposed a convex conjugate formulation:

**Theorem G.2** (Convex conjugate Rényi divergence (Birrell et al., 2023)). *Let $P, Q$ be probability distributions supported on $\Omega$, with $P \ll Q$, and let $\mathcal{M}_b(\Omega)$ denote the space of bounded measurable functions. Then, for all $\alpha \in (0, +\infty) \setminus \{1\}$,*

$$\frac{D_\alpha(P\|Q)}{\alpha} = \sup_{g \in \mathcal{M}_b(\Omega),\, g<0} \int g \, dQ + \frac{1}{\alpha - 1} \int |g|^{\frac{\alpha-1}{\alpha}} \, dP + \frac{1}{\alpha}(\log \alpha + 1). \tag{19}$$

This convex conjugate formulation removes the exponential dependence and provides more stable numerical estimates, making it preferable for our setting.

**Neural network parameterization.** To approximate $\Phi = \{g \in \mathcal{M}(\Theta) : g < 0\}$ we use the class $g_\theta$ of two-layer MLPs with spectral normalization (Miyato et al., 2018), LeakyReLU activations, and a polysoftplus output activation as in Birrell et al. (2023). The polysoftplus activation offers superior numerical stability compared to ReLU. It is defined as

$$\text{polysoftplus}(x) = -\left( \frac{1}{1-x} \mathbf{1}_{x<0} + (1+x)\mathbf{1}_{x\geq 0} \right). \tag{20}$$

The discriminator network $g_\theta$ is trained to maximize the variational bound in Equation (Lemma A.2) using samples $\{\theta_i^U\}_{i=1}^N \sim \pi_U^K$ and $\{\theta_j^R\}_{j=1}^N \sim \pi_R^{T+K}$. The optimization objective becomes:

$$\max_\theta \left\{ \frac{1}{N} \sum_{j=1}^N g_\theta(\theta_j^R) + \frac{1}{\alpha-1} \frac{1}{N} \sum_{i=1}^N |g_\theta(\theta_i^U)|^{\frac{\alpha-1}{\alpha}} + \frac{1}{\alpha}(\log \alpha + 1) \right\}. \tag{21}$$

To reduce estimator variance, we repeat the discriminator training five times with different random initializations and report the average. We use a learning rate of value 0.0001 with Adam optimizer (Kingma & Ba, 2017), and train the discriminators for 30000 epochs with batch size $b = 6000$.

This procedure used $N = 30,000$ model samples, which makes it computationally intensive and better suited for theoretical validation than for large-scale empirical benchmarking. Although regularization and repeated runs alleviate variance, Rényi divergence estimation remains a statistically challenging task. Developing scalable and lower-variance estimators is therefore an important direction for future work. The parameter $\alpha = 2$ was chosen to have a stable statistical estimation of the Rényi divergence. Large values of $\alpha$ require estimating higher moments of the density ratio involved in the Rényi divergence, and yield poor estimations. Smaller values (e.g $\alpha < 2$, on the other hand, are weaker divergence measure (since the Rényi divergence is monotone in its order $\alpha$) and do not align with the context of machine unlearning. The number of unlearning steps was selected to show that unlearning with public data has a strict computational advantage over retraining. For example, in Section 5.3, we find that with a small number of unlearning steps, we achieve an accuracy that is very close to the performance of 50 steps of retraining, using only up to 15 unlearning steps.

### G.0.2. SAMPLING FROM $\pi_U^K$ AND $\pi_R^{T+K}$

We conduct experiments on the DomainNet dataset (24-class image classification) Figure 5. We choose the domain Clipart as the private data domain, which are stylized images, and Quickdraw, a collection of hand-drawn sketches as the public domain. Image embeddings are extracted using DinoV2 (Oquab et al., 2024), a self-supervised vision transformer. We specifically use vit_small_patch16_224_dino (Caron et al., 2021). All images are resized to $224 \times 224$ prior to feature extraction.

On these embeddings, we train 30,000 linear classifiers on the full dataset $D = D_{\text{pub}} \cup D_{\text{priv}}$ for $T = 20$ iterations, and subsequently fine-tune them on the retain set $D_r = D \setminus D_{\text{forget}}$ for $K \in \{1, 5, 10, 15\}$ additional iterations. This procedure yields 30,000 samples from the unlearning distribution $\pi_U^K$.

For comparison, we train another 30,000 linear classifiers directly on the retain set $D_r$ for $T + K$ iterations, producing samples from the retraining distribution $\pi_R^{T+K}$. All models are trained using the same projected noisy gradient descent (PNGD) update with noise scale $\sigma = 0.01$, learning rate $\eta = 0.001$, batch size $b = 1024$, and radius $R = 1.0$ using SGD.

To assess robustness across dataset splits, we fix the total training set size to $N_{\text{train}} = 42{,}000$, and vary the public and forget set sizes as $(|D_{\text{pub}}|, |D_{\text{forget}}|) \in \{(10{,}000, 12{,}000), (15{,}000, 7{,}000), \text{ and } (20{,}000, 2{,}000)\}$. The remaining private data in the retain set is fixed to have size 20,000. The resulting divergence estimates are reported in Figures 3a and 3b.

### G.1. Pseudo-code

---

**Algorithm 3** Rényi Divergence Estimation via Variational Representation

---

1: **Input:** Samples $\{\theta_i^R\}_{i=1}^N \sim \pi_R^{T+K}$, $\{\theta_j^U\}_{j=1}^N \sim \pi_U^K$, order $\alpha$, discriminator architecture
2: Initialize discriminator network $g_\phi$ with spectral normalization
3: **for** epoch $= 1$ to num_epochs **do**
4:     Sample minibatch from retraining samples $\{\theta_i^R\}$
5:     Sample minibatch from unlearning samples $\{\theta_j^U\}$
6:     Compute variational objective:

$$\mathcal{L} = \frac{1}{N} \sum_{i=1}^N g_\phi(\theta_i^R) + \frac{1}{\alpha - 1} \frac{1}{N} \sum_{j=1}^N |g_\phi(\theta_j^U)|^{\frac{\alpha-1}{\alpha}} + \frac{1}{\alpha}(\log \alpha + 1) \tag{22}$$

7:     Update $\phi$ to maximize $\mathcal{L}$ via gradient ascent
8: **end for**
9: **Output:** Estimated divergence $\widehat{D}_\alpha(\pi_U^K \| \pi_R^{T+K}) = \widehat{\mathcal{L}}^{1/\alpha}$

---

# H. Evaluation with U-LiRA

### H.0.1. U-LiRA details

U-LiRA, introduced by Hayes et al. (2025) as an adaptation of the LiRA membership inference attack (Carlini et al., 2021) to the unlearning setting, formalizes unlearning evaluation as a binary hypothesis test. The goal is to distinguish between two distributions over model parameters: the unlearning distribution $\pi_U^K$, obtained by training on the full dataset and subsequently applying the target unlearning algorithm to remove the influence of the forget set, and the retraining distribution $\pi_R^{T+K}$, obtained by training from scratch without the forget set. Letting $P(\theta \mid \cdot)$ denote the likelihood of observing model parameters $\theta$ under a given distribution, the Neyman–Pearson lemma (Neyman & Pearson, 1933) implies that the most powerful test for this discrimination problem is achieved by thresholding the likelihood ratio

$$\frac{P(\theta|\pi_U^K)}{P(\theta|\pi_R^{T+K})}$$

for model parameters $\theta$.

Since directly computing $P(\theta \mid \pi_U^K)$ and $P(\theta \mid \pi_R^{T+K})$ is infeasible in practice, U-LiRA employs a series of approximations. First, the two distributions are approximated empirically by sampling: the adversary trains $N$ models under $\pi_U^K$ (full training followed by unlearning) and $N$ models under $\pi_R^{T+K}$ (training from scratch without the forget set).

To reduce the sample complexity required for a low-variance estimate, U-LiRA projects models into a one-dimensional representation space via a statistic $f : \Theta \to \mathbb{R}$ (since we only run the attack on forget sets of size 1, we follow Hayes et al. (2025) and choose $f$ to be the model's confidence score on the forget example, rescaled by the logit function $\phi(\omega) = \ln\left(\frac{\omega}{1-\omega}\right)$). The test is then conducted on the surrogate likelihood ratio

$$\frac{P(f(\theta)|f(\pi_U^K))}{P(f(\theta)|f(\pi_R^{T+K}))}.$$

As a final simplifying approximation, U-LiRA models the projected distributions as Gaussians

$$f(\pi_U^K) \approx \mathcal{N}(\mu_U, \sigma_U^2), \quad f(\pi_R^{T+K}) \approx \mathcal{N}(\mu_R \sigma_R^2),$$

where the parameters $(\mu_U, \sigma_U^2)$ and $(\mu_R, \sigma_R^2)$ are estimated directly from the $N$ sample models of each distribution.

In 5.3, we presented the attack through the lens of Bayes' rule (following Algorithm 1 of Hayes et al. (2025)), providing a more intuitive explanation for readers less familiar with hypothesis testing concepts.

### H.0.2. EXPERIMENTAL SETUP

We evaluate unlearning in binary sentiment classification of IMDB reviews (Maas et al., 2011), with Amazon product reviews (Zhang et al., 2015) as public data. Models are 2-layer LSTMs (Hochreiter & Schmidhuber, 1997), trained to minimize cross-entropy loss with projected noisy gradient descent (Gaussian noise variance $\sigma^2 = 0.01$, projection onto an $\ell_2$ ball of radius 100).

For each trial, the forget set consists of a 100 datapoints sampled uniformly from the IMDB reviews dataset. Following the U-LiRA framework, we generate 75 model samples from two distributions:

- **Unlearning distribution** $\pi_U^K$: models trained on 25,000 private datapoints plus the forget set for $T$ epochs, then finetuned without the forget set for $K$ epochs.

- **Retraining distribution** $\pi_R^{T+K}$: models trained from scratch on the same 25,000 private datapoints (excluding the forget set) for $T + K$ epochs.

We repeat this sampling process both with and without the inclusion of the 50,000 public datapoints during training and unlearning.

