# OpenReview forum: "Unlearning with Asymmetric Sources: Improved Unlearning-Utility Trade-off with Public Data"
_ICML.cc/2026/Conference — ICML 2026 regular_

### Official Review · Reviewer_18y4 · 2026-03-04

**Soundness:** 4
**Presentation:** 3
**Significance:** 3
**Originality:** 4
**Overall Recommendation:** 5
**Confidence:** 4

**Summary:**

The paper introduces Asymmetric Langevin Unlearning (ALU), a framework that uses public data to mitigate privacy costs. It proves that public data injection suppresses the unlearning cost by a factor of $O(1/n^2_{pub})$, guaranteeing a strict computational advantage over
retraining. This allows practitioners to mitigate the privacy and utility loss due to the high noise required for the unlearning impact by
increasing the volume of public data. The paper conducted extensive theoretical and experimental analyses to shed light on various aspects affecting ALU and to demonstrate its advantages in different experimental setups.

**Compliance With Llm Reviewing Policy:**

Affirmed.

**Final Justification:**

I keep my accept score.

**Key Questions For Authors:**

- I have a question about how the noise should be injected? Would it only be added to the gradient from the private dataset or also in the gradient from the public dataset? If it's not added to the gradient derived from the public dataset, and those gradients are not random variables, then how will it help reduce the impact of privacy noise?
-  If the public dataset is required to be unlearned, which is more probable since you cannot touch other private data, then would your analysis hold? If not, then how should you adapt your analysis to such cases?

**Limitations:**

Please see the weaknesess

**Strengths And Weaknesses:**

Strengths:
- The paper conducts a thorough theoretical analysis of the proposed method.
- The theoretical analysis effectively expands the existing works, allowing practitioners to mitigate the trade-off between privacy and utility.
- The paper is well written and easy to follow.
- The theoretical analysis is validated through experimental results.

Weaknesses:
- The experiments only consider small models. The paper would be stronger if you evaluated on state-of-the-art models and more advanced models, such as LLMs.
- The assumption that the public dataset will not be removed is strong, and it seems like the paper does not consider when the unlearning part is from the public dataset.

---

> ### Author Rebuttal · Authors · 2026-03-31
>
> We thank the reviewer for their positive assessment of our work.  Please find below the specific concerns and questions raised:
>
> **Evaluation on LLMs:**
>
> We agree that evaluating ALU on state-of-the-art models like LLMs would be highly valuable. However, providing a rigorous empirical evaluation of unlearning quality is a significant computational challenge. As detailed in our methodology, the U-LiRA membership inference attacks require training hundreds of model samples to achieve results with low variance. Furthermore, variational Rényi divergence estimation has a sample complexity that scales exponentially with the parameter dimension. While evaluating this on LLM-scale parameters exceeds our current compute budget, our foundational theory and medium-scale experiments strongly indicate the framework's scalability.
>
> **Assumption of Public Data Permanence:**
>
> The core of our framework is the distinction between sensitive private data and a permanent corpus of public data not subject to retraction. This "mixed-privacy" setting is well-established in current literature (Section 3.1). For analytical purposes, any data point that could potentially be subject to a future unlearning request must be treated as part of the private dataset ($D_{priv}$), regardless of its source. We briefly discuss the implications of non-permanent public data in Section 6, noting that this is an area of active research (e.g., Tramèr et al., 2024).
>
> **Q1: Noise Injection Mechanism:**
>
>  In our framework, noise is added to the aggregated gradient of the entire dataset. Following the Projected Noisy Gradient Descent (PNGD) update rule in Equation 1, the update is:
> $$\theta_{t+1} = \Pi_{\Theta} \left[ \theta_t - \eta \left( \frac{1}{n_{pub} + n_{priv}} \left( \sum_{x \in D_{pub}} \nabla l(\theta_t, x) + \sum_{x \in D_{priv}} \nabla l(\theta_t, x) \right) \right) + \xi_t \right]$$
> The presence of public data gradients reduces the impact of privacy noise by decreasing the squared sensitivity of the updates by a factor of $(n_{pub} + n_{priv})^2$. This allows us to achieve the same certifiable unlearning guarantees with a smaller noise magnitude ($\sigma^2$), thereby preserving significantly more model utility. For more clarity, we now added a pseudo-code describing the ALU procedure.
>
> **Q2: Unlearning Public Data :**
>
> We define public data as data that—by legal or procedural framework—will never be subject to unlearning requests, such as standard open-source benchmarks (e.g., ImageNet). If a practitioner faces a scenario where "public" data might be retracted, the current analysis could be adapted using frameworks like Privacy Amplification by Iteration (Feldman et al., 2019). This would provide individual privacy guarantees based on the specific iteration at which a data point was observed. This is a very promising direction for future research.

---

> > ### Author Rebuttal · Reviewer_18y4 · 2026-04-01
> >
> > The author addressed my concerns and questions. I maintain my **Accept** score.

---

### Official Review · Reviewer_496W · 2026-03-06

**Soundness:** 3
**Presentation:** 2
**Significance:** 2
**Originality:** 2
**Overall Recommendation:** 3
**Confidence:** 3

**Summary:**

This paper proposed Asymmetric Langevin Unlearning (ALU) framework by utilizing public data to mitigate privacy costs. Theoretical analysis demonstrates that ALU improves the unlearning-utility trade-off by enabling control over unlearning guarantees through data supplementation rather than noise amplification. Evaluations on image classification show the performance of the proposed framework.

**Compliance With Llm Reviewing Policy:**

Affirmed.

**Key Questions For Authors:**

See weakness above.

**Limitations:**

No. You should list the limitations of the unlearning methods proposed in this paper when handling different types of unlearning requests, such as those involving multiple categories or a small number of samples. Additionally, you should analyze the robustness of ALU when faced with malicious unlearning requests.

**Strengths And Weaknesses:**

Strengths

1.This paper demonstrates that incorporating public data establishes a more advantageous starting point for the unlearning process, thereby lowering the cost associated with unlearning.

2.This paper relaxes the idealized assumption of identical distributions and derives a new generalization bound that explicitly quantifies the trade-off between noise reduction and distribution mismatch.

Weaknesses

1.Strong assumptions: The method's reliance on leveraging knowledge from a public dataset constitutes a strong assumption. Furthermore, a distribution shift between public and private data is commonly observed. If the discrepancy between these distributions is substantial, forcibly using public data as an anchor could steer the model in a suboptimal direction.

2.Unclear contributions: The paper does not sufficiently emphasize the aspect of asymmetry or the advantages of using public data as a mechanism to improve the privacy-utility trade-off.

3.Unfair Baselines: Comparing the proposed method only against retraining in terms of time overhead is not entirely fair, as approximate unlearning methods are unlikely to achieve the exact unlearning efficacy of retraining. The comparison should include other state-of-the-art approximate unlearning methods.

4.Unsubstantiated claims: The paper claims "robustness to large-scale deletion," yet the experiments do not include evaluations specifically targeting large-scale forgetting scenarios.

5.Missing SOTA comparisons: The experimental section lacks a performance comparison with current SOTA methods.

6.Insufficient ablation and parameter analysis: The rationale behind various experimental settings is unclear. The choice of specific parameters is not justified, and there is a lack of ablation studies or sensitivity analysis for these parameters. Additionally, the criteria for partitioning public and private data, as well as the reasoning behind this split, are not explained.

7.Unlearning Metrics: The validation metric for unlearning efficacy, particularly the use of loss (e.g., in Table 1) as the primary indicator, appears to lack rigor.

---

> ### Author Rebuttal · Authors · 2026-03-31
>
> We thank the reviewer for the thoughtful feedback and for the opportunity to clarify the scope and contributions of our work. We are glad you found our relaxation of idealized identical distributions and the resulting generalization bound to be strengths of the paper. Please find below answers to the concerns expressed:
>
> **Distribution shift causing suboptimal steering:**
> You raise a valid concern that anchoring to a public dataset with a substantial distribution shift could steer the model suboptimally. This exact tension is the primary motivation behind Theorem 4.1. ALU does not assume identical distributions; rather, Theorem 4.1 explicitly quantifies the penalty of this distribution mismatch via the $D_\infty(P_{priv} || P_{pub})$ term. If the shift is too severe, our theory correctly predicts that the mismatch penalty will outweigh the noise-reduction benefits. Empirically, however, our ablation in Section 5.1 and the accuracies in Table 2 demonstrate that ALU degrades gracefully rather than failing catastrophically, even when subjected to an intentional, severe distribution shift (flipping 40% of labels).
>
>
> **Missing SOTA baselines & “unfair” comparison to retraining:**
>
> We appreciate this critique. Because ALU is an extension of Langevin Unlearning (LU) with the goal to mitigate unlearning-related “tax” on model utility, symmetric LU is our primary baseline. Comparing against exact retraining was intended to show that we can approximate the “gold standard” efficiently. However, we agree that a broader comparison could provide a more comprehensive view of the landscape. Since the core contribution of our study is to evaluate the specific advantage of incorporating public data to provide theoretical unlearning guarantees, we believe any baseline comparison should ideally be situated within this context.
> Could the reviewer clarify if they have particular state-of-the-art approximate unlearning methods in mind that are suited for studying the advantage of adding public data in providing theoretical unlearning guarantees? We are very willing to run additional experiments and compare ALU against a couple of specific methods if the reviewer has recommendations they feel are most suitable for this setting.
>
>
>
> **Unsubstantiated "mass unlearning" claims:**
>
> We apologize if our evaluation of large-scale deletion was not prominent enough. By evaluating unlearning fractions ranging from 10% to 60% of the private dataset (Figure 3.a), we are operating directly in the “mass unlearning” regime. Standard noise-based methods typically suffer catastrophic utility loss at these fractions because their required noise scales poorly. ALU's ability to maintain utility here is the empirical realization of Corollary 3.1. We provided a clearer description of these experiments as our “Mass Unlearning Evaluation” to ensure this contribution is clear.
>
> **Unlearning metrics (use of loss vs. accuracy):**
>
> We report the loss in Table 1 because it offers a granular comparison of distributional alignment, directly corresponding to the expected loss terms in our theoretical bounds (Theorem 3.4). However, we completely agree that accuracy is the standard practical metric, which is why we focus on accuracy in Table 2. Here we demonstrate that the impact on standard classification performance is indeed negligible.
>
> **Experimental settings and parameter choices:**
> - Appendix G and H provide comprehensive details on our experimental protocol and parameter selection:
> - The parameter $\alpha=2$ was chosen to have a stable statistical estimation of the Rényi divergence. Large values of $\alpha$ require estimating higher moments of the density ratio involved in the Rényi divergence, and yield poor estimations. Smaller values (e.g $\alpha < 2$, on the other hand, are weaker divergence measure (since the Rényi divergence is monotone in its order $\alpha$) and do not align with the context of machine unlearning.
> - The number of unlearning steps was selected to show that unlearning with public data has a strict computational advantage over retraining. For example, in Section 5.3, we find that with a small number of unlearning steps, we achieve an accuracy that is very close to the performance of 50 steps of retraining, using only up to 15 unlearning steps.
>
> We will update the manuscript to make the justification for these iterations more explicit. We welcome further clarification from the reviewer if there are specific additional parameters they believe require further ablation or sensitivity analysis.

---

> > ### Author Rebuttal · Reviewer_496W · 2026-04-03
> >
> > Thank you for the authors' response, which addressed some of my concerns. However, several questions remain.
> >
> > While I maintain that a thorough literature review and the selection of relevant baselines is generally the responsibility of the authors, I appreciate their openness. I have listed some recent works of unlearning. I am not insisting that the authors fully reproduce these methods; rather, an analytical comparison or discussion against SOTA approaches would better highlight the contributions of this work.
> >
> > Loss reduction does not equal successful unlearning — it may only reflect better overall data fitting. True unlearning requires that the model no longer retains specific information about deleted samples, which should be verified via membership inference attacks, extraction attacks, or statistical tests.
> >
> > A more fine-grained sensitivity analysis would be appreciated. Specifically, examining the robustness of ALU for large-scale data deletion — including details on the composition and class distribution of the deleted data — would substantially enhance the credibility of the authors' claims regarding robustness to mass unlearning. If the authors could supplement such an analysis, it would greatly strengthen the paper.
> >
> > [1] Prototype Surgery: Tailoring Neural Prototypes via Soft Labels for Efficient Machine Unlearning. CCS 2025
> >
> > [2] Towards Source-Free Machine Unlearning. CVPR 2025
> >
> > [3] Towards Scalable Exact Machine Unlearning Using Parameter-Efficient Fine-Tuning. ICLR 2025
> >
> > [4] A Certified Unlearning Approach without Access to Source Data. ICML 2025

---

> > > ### Author Response · Authors · 2026-04-05
> > >
> > > We thank the reviewer very much for the clarification. This provides more clarity into the expectations, and is a valid concern that we have addressed as follows:
> > >
> > > **1. Comparison or discussion against SOTA approaches:**
> > >  We will add the following paragraph that situates our method within the machine unlearning literature:
> > >
> > > "Recent advancements in machine unlearning have explored many strategies to mitigate the computational burden of retraining. Some methods prioritize efficiency by exclusively modifying or isolating specific components of the model, such as the final layers or via parameter-efficient fine-tuning modules (e.g., [1,3]). While effective in practice, these approaches either lack formal theoretical guarantees or require fundamental alterations to standard training pipelines and architectures, akin to SISA [5]. Another line of work investigates unlearning in restricted settings, such as when the source data is entirely inaccessible. However, these solutions rely on deterministic algorithms that assume model convergence [2], or impose strict assumptions such as strongly convex loss functions and Hessian Lipschitz continuity [4].
> > >
> > > In contrast to these paradigms, our work operates within the randomized frameworks of approximate unlearning, namely $(\epsilon, \delta)$-unlearning and Rényi unlearning, where schemes like noisy fine-tuning on the retain set have been extensively studied [6,7].
> > > We investigate specifically the unexplored benefits of incorporating public data, in the general framework of Langevin Unlearning. We highlight that this is not an algorithmic contribution, but a characterization of how the unlearning utility trade-off is improved by the asymmetric nature of this setting."
> > >
> > > **2. Clarifying Unlearning Metrics:** We concur with the reviewer's insight. We would like to clarify that in our framework, loss and accuracy are solely used to quantify how much utility is preserved after unlearning, rather than measuring the success of the unlearning itself. To evaluate actual unlearning success, we performed the exact types of rigorous checks the reviewer suggests: variational estimation of the Rényi divergence (Figure 3, Section 5.1) and extraction/membership inference attacks using U-LiRA [8] (Figure 4, Section 5.3). Both tests confirm the advantage of adding public data. We have updated the text and captions to clearly separate utility evaluation from unlearning verification.
> > >
> > > **3. Large-Scale Deletion Sensitivity Analysis:** We appreciate this suggestion, as it allows us to highlight a key strength of our paper. Figure 3a actually presents the exact fine-grained sensitivity analysis the reviewer is asking for, though we realize we did not label the deletion scales clearly enough. The graph plots three distinct scenarios: unlearning 6% (2,000 points), 21% (7,000 points), and a massive 37.5% (12,000 points) of the private dataset. Deleting over a third of the private data demonstrates our method's robustness to large-scale mass unlearning. Additionally, Tables 1 and 2 provide a secondary sensitivity analysis exploring data composition by investigating utility alterations under distribution shifts. We will update the captions and text to make these stress-tests much more explicit.
> > >
> > >
> > >
> > >
> > > [1] Prototype Surgery: Tailoring Neural Prototypes via Soft Labels for Efficient Machine Unlearning. CCS 2025
> > >
> > > [2] Towards Source-Free Machine Unlearning. CVPR 2025
> > >
> > > [3] Towards Scalable Exact Machine Unlearning Using Parameter-Efficient Fine-Tuning. ICLR 2025
> > >
> > > [4] A Certified Unlearning Approach without Access to Source Data. ICML 2025
> > >
> > > [5] Machine unlearning. IEEE, 2021.
> > >
> > > [6] Langevin unlearning: A new perspective of noisy gradient descent for machine unlearning. Neurips 2024
> > >
> > > [7] Certified Unlearning for Neural Networks. ICML 2025
> > >
> > > [8] Inexact Unlearning Needs More Careful Evaluations to Avoid a False Sense of Privacy. SaTML 2025

---

### Official Review · Reviewer_6qpn · 2026-03-08

**Soundness:** 4
**Presentation:** 3
**Significance:** 3
**Originality:** 3
**Overall Recommendation:** 5
**Confidence:** 4

**Summary:**

This paper asks an intriguing question: can using public data help improve the balance between privacy and utility in machine unlearning?

This is especially relevant for Langevin-style methods, which often struggle to provide practical guarantees when there are many deletions.

The main idea is that public data serves as a stabilizing anchor, making the original training and retraining distributions more similar and reducing the amount of noise needed. The paper explores how this effect depends not only on the amount of public data, but also on how well it matches the private data.
The paper supports this with a theoretical analysis based on Rényi divergence and log-Sobolev tools, plus experiments using divergence estimation and membership-inference-style evaluations.

**Compliance With Llm Reviewing Policy:**

Affirmed.

**Key Questions For Authors:**

1. The paper still seems to rely on strong-convexity assumptions. How should readers interpret “strict computational advantage over retraining” in the context of the actual deep learning experiments?

2. Can you separate the effect of using public data as an anchor from other simpler explanations, such as just increasing the total amount of training data or improving optimization stability? An ablation study with the same total data but different public/private splits would help clarify this.

**Strengths And Weaknesses:**

1. The paper clearly and timely defines the problem: the noise required for deletion often severely reduces model utility. The motivation is strong, and bringing public data into the unlearning analysis is a fresh idea.

2. Theoretical structure is coherent.
- they bound the mismatch between the original learning distribution and the retraining distribution;
- they show that unlearning converges to retraining;
- and they analyze the resulting utility.

Theorem 4.1 is especially valuable because it highlights a penalty for mismatches, helping to avoid an overly idealized version of the problem.

---

> ### Author Rebuttal · Authors · 2026-03-31
>
> We sincerely thank the reviewer for their thoughtful and positive feedback, and recognizing the coherence of our theoretical contributions.
>
> **Q1: Computational advantage in the non-convex setting:**
>
> You raise an excellent question regarding the interpretation of “strict computational advantage” without strong convexity.
> As discussed in Appendix A.2, Log-Sobolev assumptions involve iteration-dependent constants that are often intractable, making it difficult to precisely characterize when unlearning becomes less costly than retraining in non-convex landscapes. However, the core mechanism still holds: our results show that the number of unlearning steps required to reach a target divergence decreases quadratically with the number of public points ($n_{pub}$). Therefore, even in non-convex regime, increasing $n_{pub}$ eventually makes unlearning computationally more advantageous than retraining, even if the exact rate at which it does is obscured by the intractability of the constants.
> We now added a clarifying remark in the main text to explicitly bridge our theoretical convex bounds with non-convex practical expectations.
>
> **Q2: Separating the public data effect from total data volume:**
>
> We completely agree that isolating the “asymmetric anchor” effect from the stabilizing effect of “more data” is critical. To address this, we point to Figure 3a, which exactly represents the ablation study you suggested. In this experiment, the total number of training points is held strictly constant, while the split between public and private data is varied. The results demonstrate that injecting public data directly reduces the divergence, independent of total data volume. We realize our original caption did not make this constant-volume constraint obvious. We now updated it in the revision to clearly highlight this isolation of the asymmetric effect.

---

> > ### Author Rebuttal · Reviewer_6qpn · 2026-04-03
> >
> > Thanks for addressing my concerns. I keep the accept score.

---

### Decision · Program_Chairs · 2026-04-30

**Decision:**

Accept (regular)

**Comment:**

This paper studies machine unlearning with public data, introducing Asymmetric Langevin Unlearning to improve the unlearning–utility trade-off. Reviewers agreed that the problem is timely and that leveraging public data as an “anchor” is a new and well-motivated idea.

The paper provides a coherent theoretical analysis showing how public data reduces divergence between training and retraining distributions, including an explicit treatment of distribution mismatch. The empirical results support these claims, and the rebuttal clarified some key points, including isolating the effect of public data from simply increasing dataset size and strengthening the evaluation of unlearning efficacy.

The main concerns relate to assumptions (e.g., availability of public data), presentation, and the breadth of experimental comparisons. However, these are largely issues of scope and exposition rather than fundamental flaws.

Overall, the paper makes a clear and useful contribution to an important problem, and I recommend acceptance, with the expectation that the final version makes some suggested improvements in the presentation.